# Dependency-aware Maximum Likelihood Estimation for Active Learning

**Beyza Kalkanlı**                                                              *kalkanli@ece.neu.edu*
*Department of Electrical and Computer Engineering*
*Northeastern University*

**Tales Imbiriba**                                                           *tales.imbiriba@umb.edu*
*Department of Computer Science*
*University of Massachusetts Boston*

**Stratis Ioannidis**                                                         *ioannidis@ece.neu.edu*
*Department of Electrical and Computer Engineering*
*Northeastern University*

**Deniz Erdoğmuş**                                                           *erdogmus@ece.neu.edu*
*Department of Electrical and Computer Engineering*
*Northeastern University*

**Jennifer Dy**                                                                   *jdy@ece.neu.edu*
*Department of Electrical and Computer Engineering*
*Northeastern University*

**Reviewed on OpenReview:** *https://openreview.net/forum?id=qDVDSXXGK1*

## Abstract

Active learning aims to efficiently build a labeled training set by strategically selecting samples to query labels from annotators. In this sequential process, each sample acquisition influences subsequent selections, causing dependencies among samples in the labeled set. However, these dependencies are overlooked during the model parameter estimation stage when updating the model using Maximum Likelihood Estimation (MLE), a conventional method that assumes independent and identically distributed (i.i.d.) data. We propose Dependency-aware MLE (DMLE), which corrects MLE within the active learning framework by addressing sample dependencies typically neglected due to the i.i.d. assumption, ensuring consistency with active learning principles in the model parameter estimation process. This improved method achieves superior performance across multiple benchmark datasets, reaching higher performance in earlier cycles compared to conventional MLE. Specifically, we observe average accuracy improvements of 6%, 8.6%, and 10.5% for $k = 1$, $k = 5$, and $k = 10$ respectively, after collecting the first 100 samples, where entropy is the acquisition function and $k$ is the query batch size acquired at every active learning cycle.

## 1 Introduction

Supervised learning methods rely on the availability of abundant labeled data (Alzubaidi et al., 2023). The significance of large labeled sets becomes more acute with over-parameterized deep learning models that can be prone to overfitting (Brigato & Iocchi, 2021). Resource constraints in acquiring labeled data present a practical challenge and must be balanced against the benefits of having more labeled data (Alzubaidi et al., 2023; Bansal et al., 2022). In particular, the costs associated with labeling motivate the need to use labeling resources as effectively as possible. Data augmentation (Shorten & Khoshgoftaar, 2019) and transfer learning methods (Tan et al., 2018; Yu et al., 2022) have been adopted as methods to deal with this (labeled) data

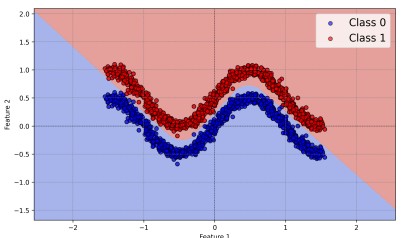

Passive learning with IMLE (700 samples)

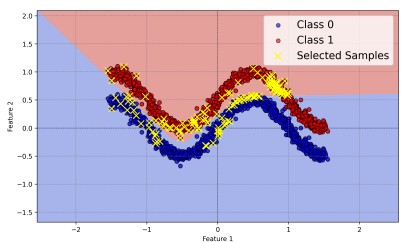

Active learning with IMLE and entropy as an acquisition function (140 samples)

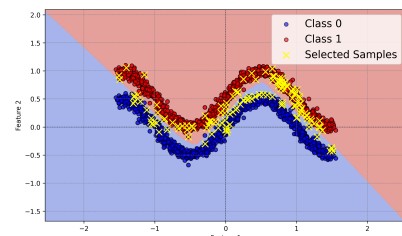

Active learning with DMLE and entropy as an acquisition function (140 samples)

Figure 1: Comparison of decision boundaries and selected samples across different learning scenarios: On the left, all 700 samples are used for model training without active learning. In the active learning experiments shown in the middle and right, the entropy acquisition function is used for sample selection, with one sample acquired at each cycle, resulting in a total of 140 samples. The middle setup employs IMLE for model parameter updates, while the right setup uses DMLE. The shaded regions represent the decision boundaries, and the yellow crosses in the active learning setups highlight the selected informative samples. Both the model trained with all samples using IMLE and the active learning model updated with DMLE achieve 99.5% accuracy, while the active learning model using IMLE for parameter updates achieves only 92% accuracy.

scarcity challenge. In contrast to these methods, active learning aims to intentionally select which samples to acquire or labels to request based on the latest available model, attempting to achieve a high return on performance improvement per data acquisition or labeling resource spent (Ren et al., 2021).

Active learning is an iterative process that alternates between two steps: (1) querying samples/labels and (2) adapting or retraining the model with the updated training set (Settles, 2009). Each complete iteration of these steps is referred to as a *cycle*. In each cycle, the model might either acquire one sample or a *batch* of samples during the query sampling step. So, in an active learning setup, previously selected data have control over the selection of new samples at each cycle through the updated model parameters. This sequential nature of data collection in active learning inherently introduces dependencies between samples across cycles in the labeled set. However, conventional Maximum Likelihood Estimation (MLE) assumes samples are i.i.d., overlooking dependencies during model parameter estimation; hence, we refer to it as Independent MLE (IMLE).

Sample dependency has been studied in the literature by revisiting the sample acquisition stage (Sener & Savarese, 2018; Ash et al., 2019a) or re-weighting the objective function (Farquhar et al., 2021; Beygelzimer et al., 2008), however, these approaches are insufficient to capture the dependency within the objective function, as they still rely on the i.i.d. assumption of MLE. Neglecting sample dependency during model parameter estimation is fundamentally incompatible with the nature of active learning. Ignoring this dependency during training leads to a vicious cycle of suboptimal model parameter estimation and, in turn, poor sample acquisition decisions in upcoming cycles as illustrated in Figure 1. The model updated with DMLE, using the same number of samples, outperforms the one updated with IMLE, creating a decision boundary similar to the one achieved by using all data samples with passive learning. The diverse and well-separated sample selections made by the DMLE-updated model show that a better model leads to more effective sample selection and better results with fewer samples. This motivates us to explore ways to model this dependency in the training objective. To the best of our knowledge, this is the first work that attempts to correct for dependencies among samples across active learning cycles during model parameter estimation by removing the incompatible i.i.d. data assumption from the training objective. Our main contributions are as follows:

- We propose Dependency-aware Maximum Likelihood Estimation (DMLE), a novel approach that corrects MLE by eliminating the i.i.d. data assumption and aligning with active learning principles through explicit modeling of sample dependencies during the parameter estimation stage.

- We theoretically derive the dependency term in MLE that arises from the influence of previously selected samples on subsequent sample selections within the active learning framework.

- Our empirical results and hypothesis tests indicate that, in an active learning process, DMLE outperforms the conventional independent MLE (IMLE) approach in various benchmark datasets and several sample selection strategies.

The remainder of this paper is organized as follows. The related works are discussed in Section 2. Section 3 provides the necessary definitions and background in active learning. We present the proposed methodology of DMLE to use MLE within active learning in Section 4. Experiments and results are presented in Section 5 while final remarks are made in Section 6.

## 2 Related Work

In machine learning, the dataset used for training plays a significant role in resulting model performance (Shui et al., 2020; Gal et al., 2017). However, labeling large datasets is not always convenient and straightforward, especially in domains such as medicine (Hoi et al., 2006; Smailagic et al., 2018; Budd et al., 2021), hyperspectral imaging (Cao et al., 2020; Lei et al., 2022; Thoreau et al., 2022), or bioinformatics (Mohamed et al., 2010; De Angeli et al., 2021). In the active learning paradigm, the model is allowed to choose the data from which it learns, aiming to enhance labeling efficiency (Settles, 2009). Thus, using active learning becomes crucial to create the best training set under limited resources. For the optimization of this sample selection procedure, extensive research in active learning has been focused on developing various acquisition strategies (Schohn & Cohn, 2000; Zheng & Padmanabhan, 2002; Sourati et al., 2017; Ash et al., 2021; Kim et al., 2023) aiming to make better selections at each cycle.

Initial work in active learning aimed to combine this paradigm with simple machine learning models, such as Support Vector Machine (SVM), decision tree classifier (Lewis & Catlett, 1994a), nearest neighbor classifier (Lindenbaum et al., 2004), and logistic regression model (Schein & Ungar, 2007; Zhang & Oles, 2000). With neural networks' increasing popularity, the deep active learning research focused on combining neural networks with active learning has become accelerated (Wang et al., 2021; Ban et al., 2023). Despite the merits of deep active learning, Settles (2011) draws attention to the sample correlation problem that emerges in batch sample acquisition. While batch selection may reduce the overall computational expenses of training after each cycle, it can lead to redundant labeling if the trade-off between uncertainty and diversity of samples is not well-balanced, potentially hindering the objectives of active learning (Settles, 2011; Krishnan et al., 2021). To solve this issue, various corrective approaches have been proposed mainly attempting to balance inference uncertainty and sample diversity during batch sample selection (Sener & Savarese, 2018; Ash et al., 2019a; Kirsch et al., 2019b; Ash et al., 2019b; Bıyık et al., 2019; Citovsky et al., 2021; Kirsch et al., 2019a). Kirsch et al. (2021) highlights computational expenses of methods aiming to combine uncertainty and diversity of samples and proposes a stochastic batch sampling strategy with less additional costs during the sample selection. Although sample correlations are comprehensively researched, the focus has so far been on the dependency between the batch of samples selected in a cycle, primarily addressing the sample dependency within the sample acquisition stage. Our aim differs from the earlier works as we emphasize sample dependency across cycles and address it during the model parameter estimation stage.

Achieving good results at earlier cycles even when starting from a few or zero labeled samples is highly preferable in active learning (Jin et al., 2022; Yuan et al., 2020; Houlsby et al., 2014). Challenges include dealing with biased data (Gudovskiy et al., 2020; Singh et al., 2023), the creation of imbalanced training data across the cycles (Kottke et al., 2021; Yang & Loog, 2022; Szűcs & Papp, 2022), the selection of outliers (Chen et al., 2022; Beygelzimer et al., 2008; Prabhu et al., 2019; Dasgupta & Hsu, 2008), or sample dependency in data collection leading to a possible discrepancy between the acquired training data distribution and the actual population distribution (Dasgupta, 2011). Among the listed challenges, the one most aligned with our focus is the emphasis on sample dependency. Previous work on this issue has proposed using re-weighting the objective function (Farquhar et al., 2021; Beygelzimer et al., 2008) or clustering techniques (Dasgupta & Hsu, 2008) to address sample dependency, with a focus on achieving a representative approximation of the true distribution. While Dasgupta & Hsu (2008) proposes a clustering-based sample selection process

aimed at ensuring good representation of the population with the selected sample set, their approach is limited to optimizing the sample acquisition stage with a similar goal to that of batch sample selection methods. Beygelzimer et al. (2008) on the other hand, proposes an importance-weighted active learning method to address the sample dependency relationship by applying importance-sampling corrections in a stream-based active learning scenario, which is limited to simple binary classifier models. Although Farquhar et al. (2021) also focuses on neural networks and addresses an unlabeled pool scenario similar to ours, their method is limited to re-weighting within the objective aiming to come up with an unbiased risk estimator. They conclude that applying this methodology with neural networks leads to performance deficiencies due to the loss of generalizability. However, the sample dependency issue we highlight arises specifically during the model parameter estimation stage due to the i.i.d. data assumption inherent in MLE. The proposed methods cannot capture this issue, as fixing it by re-weighting the loss term for each sample during training will still rely on the i.i.d. assumption and neglect the dependency term (See Appendix D for experimental comparison on MNIST dataset).

## 3 Background

We consider an active learning setup (Settles, 2009; Takezoe et al., 2023; Joshi et al., 2009), in which a large number of unlabeled samples are available for an evaluation in terms of an acquisition score and labeling by an oracle (e.g., an expert) on demand. An active learning algorithm proceeds in stages, collecting a batch of $k$ labels from the labeling oracle, training a model from the labeled data, and selecting which batch to collect next. At cycle $t$, we define the unlabeled dataset as $U_t = \{x_i\}_{i=1}^{N_U^t}$ and the labeled set as $D_t = \{(x_i, y_i)\}_{i=1}^{N_L^t}$ where $x_i \in \mathcal{X} \subset \mathbb{R}^{d_x}$ are the features, $y_i \in \mathcal{Y} = \{1, \dots, K\}$ are categorical labels. We also denote by $N_U^t$ the number of unlabeled samples and $N_L^t$ the number of labeled samples at cycle $t$. We assume that, conditioned on parameters $\theta$ and sample features $x$, label $y$ is sampled from a conditional distribution $P(\cdot \mid x; \theta)$ parameterized by $\theta \in \mathbb{R}^{d_\theta}$. For example, in a deep neural-network (DNN) setting, where the DNN model is represented by function $f_\theta : \mathcal{X} \to \{0, 1\}^K$, the probability would be given by the softmax function, i.e., $P(y \mid x, \theta) = e^{(f_\theta(x))_y} / \sum_{y'} e^{(f_\theta(x))_{y'}}$, for $y, y' \in \mathcal{Y}$. For clarity, a glossary of the terminology is included in Appendix J.

Overall, the above active learning algorithm is formally defined by two components, illustrated in Figure 2: *query sampling*, which includes an *acquisition function* and a *sample selection* strategy, and a *model parameter estimation* method.

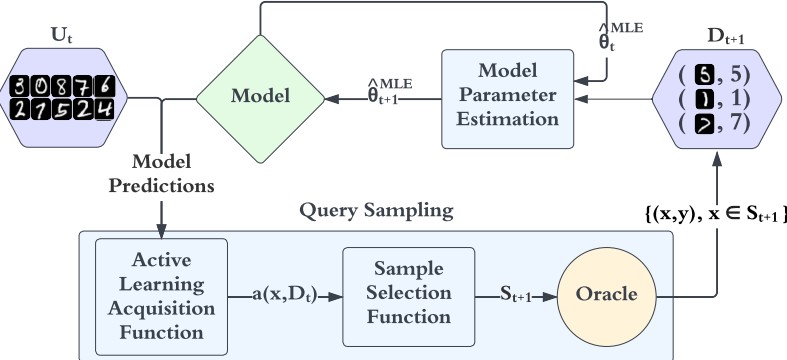

Figure 2: Components of active learning procedure. In active learning, the model selects the samples to be acquired where the model parameters are cyclically updated with the updated labeled set. First, uncertainty scores for the samples in the unlabeled sample pool $U_t$ are calculated using the current model with the acquisition function $a(x, D)$. Next, samples $S_{t+1}$ selected based on the sample selection strategy are labeled by an oracle and included in the labeled set $D_t$, resulting in $D_{t+1} = D_t \cup \{(x, y), x \in S_{t+1}\}$. After the query sampling step, the updated labeled set is used for the model update with a model parameter estimation method, concluding one cycle.

### 3.1 Acquisition Function

Acquisition functions (Lewis & Catlett, 1994b; Settles, 2009) model the value of acquiring a label for an unlabeled sample towards the quality of the trained model. Formally, the acquisition function $a : \mathcal{X} \times \mathbb{R}^{d_\theta} \to \mathbb{R}$ takes as arguments an $x \in U_t$, current labeled set $D_t$, model parameters $\theta$, and returns the value of the sample given the current parameters.

This is often modeled through the uncertainty of the label prediction of the classifier trained so far at an unlabeled sample. Examples include the entropy (Shannon, 1948), Bayesian Active Learning by Disagreement (BALD) (Houlsby et al., 2011), margin sampling (Roth & Small, 2006), and least confidence (Settles, 2009). The entropy acquisition function quantifies the uncertainty of a sample using the entropy of label $y$ as follows:

$$a(x, D_t, \theta) = \mathrm{H}[y|x, \theta; D_t] = -\sum_{y \in \mathcal{Y}} P(y \mid x; D_t, \theta) \ln P(y \mid x; D_t, \theta). \tag{1}$$

The BALD acquisition function evaluates the uncertainty of a sample as the mutual information between label predictions and model posterior:

$$a(x, D_t, \theta) = \mathrm{H}[y|x, D_t, \theta] - \mathbb{E}_{p(\theta|D_t)}[\mathrm{H}[y|x, \theta]]. \tag{2}$$

The least confidence acquisition function quantifies the uncertainty of a sample as:

$$a(x, D_t, \theta) = 1 - \max_y P(y \mid x; D_t, \theta). \tag{3}$$

Some acquisition methods extend beyond uncertainty-based selection by incorporating sample diversity, ensuring that selected samples cover a broader representation of the data. This approach is considered deterministic, as demonstrated by methods such as Core-set (Sener & Savarese, 2018), BADGE (Ash et al., 2019b), and Cluster Margin (Citovsky et al., 2021). Depending on the acquisition method, acquisition scores can be designed to align with the selection procedure. We provide an acquisition score for the Core-set method below, where $C$ is the set of cluster centers derived from $D_t$, and the feature representations extracted by the model with parameters $\theta$ are denoted as $\phi(x, \theta)$:

$$a(x, D_t, \theta) = \min_{x' \in C} ||\phi(x, \theta) - \phi(x', \theta)||_2. \tag{4}$$

### 3.2 Sample Selection Strategy

The selection strategy determines the samples from the unlabeled set to be labeled in the next cycle relying on the acquisition scores. Two of the most widely employed approaches for sample selection are Top-$k$ selection (Gal et al., 2017; Kirsch et al., 2019a) and stochastic batch sampling: proposed recently. The latter was shown to outperform Top-$k$ in selecting informative samples (Kirsch et al., 2021) with less computational expenses which makes it preferable for sample selection during cycles. Independently of the selection strategy, having selected $S_{t+1}$, the corresponding labels are collected from the labeler, yielding a dataset

$$\partial D_{t+1} = \{(x, y) : x \in S_{t+1}\}.$$

and sets $D_t$ and $U_t$ are adjusted accordingly, i.e., $D_{t+1} = D_t \cup \partial D_{t+1}$ and $U_{t+1} = U_t \setminus S_{t+1}$.

**Top-$k$ Selection.** In Top-$k$ selection procedure, the selected batch of size $k$ is given by:

$$S_{t+1} = \underset{S \subset U_t : |S| = k}{\arg\max} \sum_{x \in S} a(x, D_t, \theta), \tag{5}$$

where $D_t$ is the labeled set at cycle $t$ and $U_t$ is the unlabeled set. As the objective function in equation 5 is submodular, it is maximized by the items of the highest acquisition value (Kirsch et al., 2021).

Table 1: Possible sample selection probability distributions spanned by the sampling method of Kirsch et al. (2021). Here, $a(x, D, \theta)$ represents the acquisition score, $r$ is the descending ranking of the acquisition scores $a(x, D, \theta)$, with the smallest rank as 1, and $\beta$ denotes the coldness parameter.

| Distribution | $P(x|D_t; \theta)$ |
|---|---|
| Softmax | $\propto e^{\beta a(x, D_t, \theta)}$ |
| Power | $\propto a(x, D_t, \theta)^{\beta}$ |
| Soft-rank | $\propto r^{-\beta}$ |

**Stochastic Batch Selection.** For stochastic batch selection, Kirsch et al. (2021) propose three probabilistic sampling schemes: Softmax, Power, and Soft-Rank Acquisition, as listed in Table 1. Stochastic batch selection methods differ from the conventional Top-$k$ selection method by perturbing the acquisition scores/ranks of the samples in the unlabeled set with Gumbel noise prior to the Top-$k$ acquisition, formulated as follows:

$$S_{t+1} = \underset{S \subset U_t : |S| = k}{\arg\max} \sum_{x \in S} \left[ \tilde{a}(x, D_t, \theta) + \epsilon \right]. \tag{6}$$

where $\tilde{a}$ is a modified acquisition function and $\epsilon$ is Gumbel noise. This perturbation introduces probabilistic selection into the sample selection procedure. Kirsch et al. (2021) show that, given an acquisition function $a(\cdot)$, the sample selection distributions $P(x|D_t; \theta)$ presented on Table 1 can be implemented via equation 6, through an appropriate choice of $\tilde{a}(\cdot)$ and parameters of the Gumbel noise. Details are provided in Appendix A in the supplement.

There are two ways to incorporate clustering-based active learning methods as acquisition functions with the selection methods outlined here. The first approach involves using a measurement based on the clustering/diversity method as the acquisition score, which can be used for stochastic softmax sampling, stochastic power sampling, or top-k sampling. The second approach entails developing a ranking of the samples in the unlabeled set to be used with stochastic soft-rank sampling. As long as acquisition scores or ranks can be computed, sample dependency can be integrated into model training within any active learning framework.

### 3.3 Model Parameter Estimation

In the model adaptation/re-training step, typically, Maximum Likelihood parameter Estimation (MLE) is preferred:

$$\hat{\theta}_{t+1}^{\text{MLE}} = \underset{\theta}{\arg\max} \ln P(D_{t+1}; \theta) = \underset{\theta}{\arg\max} \sum_{(x,y) \in D_{t+1}} \ln P(y \mid x; \theta) + \sum_{\tau=1}^{t+1} \ln P(S_\tau \mid D_{\tau-1}; \theta) \tag{7}$$

The foundational assumption of i.i.d. data in MLE (Bishop, 2006) causes the $\ln P(S_\tau|D_{\tau-1}; \theta)$ term to be a constant and leads to the following conventional objective function:

$$\hat{\theta}_{t+1}^{\text{MLE}} = \underset{\theta}{\arg\max} \sum_{(x,y) \in D_{t+1}} \ln P(y \mid x; \theta) \tag{8}$$

which we refer to as Independent Maximum Likelihood Estimation (IMLE) in the rest of the paper to highlight this sample independence assumption (see Appendix B for detailed derivation).

## 4 Dependency-aware Maximum Likelihood Estimation

In the iterative framework of active learning, the sample acquisition is followed by the adaptation/re-training of the model. Conventional approaches often employ MLE given in equation 8 as a way of doing model parameter estimation during this stage. Conventional MLE, referred to here as IMLE, introduces a fundamental

incompatibility in active learning: its assumption of sample independence conflicts with the sequential nature of sample acquisition. The i.i.d. data assumption in IMLE eliminates the contribution of sample dependencies during estimation (see Appendix B). However, *labels in $D_{t+1}$* are in fact *not* independent, as the data samples in the labeled set are not i.i.d.: their selection is determined via the active learning algorithm itself, which in turn depends on the past collected labels. As a result, IMLE is an approximation of reality and overlooks sample dependencies in the estimate of $\theta$. Our major contribution is to account for these sample dependencies during model parameter estimation.

Ideally, we aim to preserve the valuable information embedded in sample dependencies within the likelihood $P(D_{t+1}; \theta)$, which is typically disregarded under the i.i.d. assumption. In equation 7, if the second term is not eliminated due to the i.i.d. assumption, it gives rise to the dependency term in equation 11. This dependency appears as a conditional probability term of the form $P(S_\tau \mid D_{\tau-1}; \theta)$. This accounts for the dependence of batches on previous cycles and may enable a conditional chain computation of the likelihood $P(D_{t+1}; \theta)$ as follows:

$$\hat{\theta}_{t+1}^{DMLE} = \arg\max_\theta \sum_{\tau=1}^{t+1} \left[ \sum_{i=1}^{k} \ln P(y_{\tau,i} \mid x_{\tau,i}; \theta) + \ln P(S_\tau \mid D_{\tau-1}; \theta) \right] \tag{9}$$

$$= \arg\max_\theta \sum_{(x,y) \in D_{t+1}} \ln P(y \mid x; \theta) + \sum_{\tau=1}^{t+1} \ln P(S_\tau \mid D_{\tau-1}; \theta) \tag{10}$$

$$= \arg\max_\theta \underbrace{\sum_{(x,y) \in D_{t+1}} \ln P(y \mid x; \theta)}_{\text{IMLE}} + \underbrace{\sum_{\tau=1}^{t+1} \sum_{x \in S_\tau} \ln P(x \mid D_{\tau-1}; \theta)}_{\text{Dependency}} \tag{11}$$

We propose taking into account these dependencies as evident in equation 11, during the model parameter estimation. We refer to this approach as Dependency-aware Maximum Likelihood Estimation (DMLE), in which model parameters are updated while incorporating the sample dependencies introduced during active learning cycles.

However, this poses a challenge, as $P(S_\tau \mid D_{\tau-1}; \theta)$ distribution is difficult to express in a closed form: it depends on the labels in $D_{t+1}$ through the generation of an estimate of $\theta$, which affects the acquisition function and in turn the sample selection. Fortunately, when utilizing the stochastic batch sampling approaches introduced by Kirsch et al. (2021) and discussed in Section 3.2, an appropriate corresponding probability distribution $P(S_\tau \mid D_{\tau-1}; \theta)$ is associated with them (see Table 1). This allows us to introduce an appropriate correction term for equation 11. When using the Top-$k$ sampling discussed in Section 3.2, we approximate $P(S_\tau \mid D_{\tau-1}; \theta)$ by the Softmax distribution given in Table 1 (Bishop, 2006), with empirical support for this choice provided in Appendix I.

Unlike the formulations presented by Raginsky & Rakhlin (2011) (equation 11) and discussed in Chap. 15 of Lattimore & Szepesvári (2019), our approach explicitly models both the data likelihood and the sample selection distribution as functions of a single underlying parameter $\theta$. This choice reflects the interactive nature of active learning, where the evolving model directly influences which samples are selected, and the acquired labels in turn refine the model. By updating the selection distribution at each cycle using the most recent parameter estimates, our formulation ensures that past acquisitions are re-evaluated under the current model, capturing the feedback loop between selection and learning. This stands in contrast to approaches that treat the selection mechanism as fixed or independent of the model dynamics, which neglect the dependencies inherent to the sequential data generation process in active learning.

The comparison of DMLE in equation 11 with IMLE in equation 8 highlights their discrepancy in the latter term. We interpret this term as accounting for the sample dependencies resulting from active learning in MLE. Indeed, this term incorporates the likelihood of observed sample features (rather than labels alone) under a given $\theta$, as induced by sampling. Thus, the selection of the sampling strategy results in varying terms in the objective function, which are proportionate to the summation of acquisition scores of samples in

the labeled set, as can be deduced from Table 1 (see Appendix A for the corresponding objective functions). This likelihood function directly influences the Fisher Information matrix by incorporating the dependencies between selected samples, leading to a stronger information measure. As a result, the estimator benefits from lower variance guarantees. In Appendix C.1, we prove:

**Theorem 1** *Let $I^{IMLE}(\theta)$ and $I^{DMLE}(\theta)$ be the Fisher Information matrices for IMLE and DMLE, respectively. As DMLE accounts for sample dependencies, $I^{DMLE}(\theta) \succeq I^{IMLE}(\theta)$. By the Cramér-Rao bound, this implies $Cov(\hat{\theta}^{DMLE}) \preceq Cov(\hat{\theta}^{IMLE})$, ensuring that DMLE provides lower variance estimates compared to IMLE.*

Thus, the incorporation of sample dependencies in DMLE strengthens the estimator, leading to improved parameter precision. To better understand the impact of training with DMLE, it is helpful to examine the expected log-likelihood over samples drawn from $P^{true}$ which is the true data distribution. This provides a theoretical justification for DMLE, as it reveals how the decomposition of the log-likelihood guides the alignment of model distributions with the true data distribution. In Appendix C.2, we show:

**Theorem 2** *Let $P^{true}$ be the true data distribution, $(x', y') \sim P^{true}$, and $P(y' \mid x', D_t; \hat{\theta})$ and $P(x' \mid D_t; \hat{\theta})$ be the distributions estimated from actively selected training data $D_t$. The expected log-likelihood on this data decomposes as:*

$$\mathbb{E}_{(x', y') \sim P^{true}}\left[\ln L(\hat{\theta})\right] = -D_{KL}(P(y' \mid x') \parallel P(y' \mid x', D_t; \hat{\theta})) - D_{KL}(P(x') \parallel P(x' \mid D_t; \hat{\theta})).$$

When the training data is actively selected, DMLE, which includes both terms in the log-likelihood decomposition, minimizes the sum of two KL divergences, ensuring better alignment with the true distribution compared to IMLE, which ignores the dependency term. The dependency term, $D_{\mathrm{KL}}\left(P(x') \parallel P(x' \mid D_t; \hat{\theta})\right)$, measures the discrepancy between the true distribution $P(x')$ and the model's estimate $P(x' \mid D_t; \hat{\theta})$ learned from the actively selected data. Maximizing the expected log-likelihood is equivalent to minimizing the sum of two KL divergences, ensuring that both the predictive and data distributions of the model remain close to the true distributions. By addressing this discrepancy, DMLE reduces distributional mismatch and enhances the model's approximation of the true data distribution. In contrast, IMLE, by neglecting this term, risks a greater distributional gap. The dependency term ensures that the model's input distribution remains aligned with the true data distribution. The overall algorithm with DMLE in the active learning framework is provided in Alg. 1.

---

**Algorithm 1** Dependency-aware Maximum Likelihood Estimation for Active Learning

---

**Input:** Unlabeled dataset $U_\tau = \{x_i\}_{i=1}^{N_U^\tau}$, Labeled dataset $D_\tau = \{(x_i, y_i)\}_{i=1}^{N_L^\tau}$, Model $f_\theta$, Sample selection size $k$

**for** $\tau = 1, ..., T$ **do**

    **Acquisition Function:** Compute acquisition scores $a(x, D_\tau, \theta)$ for all $x \in U_\tau$ with $f_\theta$

    **Sample Selection Strategy:** Select batch $S_{\tau+1} \subset U_\tau$ of size $k$

    **Oracle Labeling**

    - Request labels from Oracle for samples in $S_{\tau+1}$

    - Obtain labeled samples $\{(x_i, y_i)\}_{x_i \in S_{\tau+1}}$

    - Update Labeled Dataset $D_{\tau+1} \leftarrow D_\tau \cup \{(x_i, y_i) \mid x_i \in S_{\tau+1}\}$

    - Update Unlabeled Dataset $U_{\tau+1} \leftarrow U_\tau \setminus S_{\tau+1}$

    **Model Parameter Estimation**

    - Compute acquisition scores $a(x, D_\tau, \theta)$ for all $x \in D_{\tau+1}$

    - Update model parameters $\hat{\theta}_{\tau+1}$ with DMLE (equation 11) on $D_{\tau+1}$

**end for**

---

# 5 Experiments

## 5.1 Experiment Setting

**Datasets and Models.** We evaluate the impact of including the dependency term during model parameter estimation for the active learning process on the model's prediction performance by testing it on datasets consisting of images, texts, or features; to have diverse experiments, we used datasets with different sizes and complexities. We use eight classification datasets: Iris (Fisher, 1988), Mnist (Deng, 2012), Fashion-Mnist (FMnist) (Xiao et al., 2017), SVHN (Netzer et al., 2011), Reuters-21578 (Reuters) (Lewis, 1997), Emnist-Letters (Emnist) (Cohen et al., 2017), Cifar-10 (Krizhevsky et al., 2009), and Tiny ImageNet (Le & Yang, 2015). We randomly sample a portion of each dataset and assume that only the subset was available during training. We used a subset of 30000 samples for Mnist and FMnist, 110 samples for Iris, 14651 samples for SVHN, 8083 samples for Reuters, 4440 samples for Emnist, 3000 samples for Cifar-10, and 1500 samples for Tiny ImageNet. Depending on the dataset, we adjust the complexity of the model. We used LeNet (Lecun et al., 1998) for Mnist, FMnist, SVHN, and Emnist, a two-layer MLP for Iris, a neural network with embedding and convolutional layers for Reuters, ResNet-50 (He et al., 2015) for Cifar-10, and a visual transformer (ViT-B/16) with 200 classes fine-tuned from ImageNet weights (Dosovitskiy et al., 2021) for Tiny ImageNet.

**Comparing methods.** In this paper, we address the i.i.d. data assumption in model parameter estimation, emphasizing its fundamental incompatibility with active learning. We propose the Dependency-aware Maximum Likelihood Estimation (DMLE) method, which accounts for dependencies between data points, a problem that has not been tackled in the literature before, and applies to any setup irrespective of the employed sample selection strategy or acquisition function. Thus, we compare our method with the conventional model parameter estimation technique in active learning setups, which we refer to as Independent Maximum Likelihood Estimation (IMLE). With the experiments, we assess the performance disparity between using DMLE and IMLE while utilizing different combinations of acquisition functions and sample selection strategies to show the impact of considering sample dependency during the model parameter estimation.

We use Keras for the implementation of the neural networks (Chollet et al., 2015). Aiming a fair comparison, at each active learning cycle, for all combinations, we use the same number of epochs, the same hyperparameter combinations, and the same acquisition functions. We present results with $\beta = 1$ for the stochastic batch selection as practiced and suggested by Kirsch et al. (2021) which can be important for the model and acquisition performance. Different values for $\beta$ might be explored further for better results. For all models, we use Adam optimizer (Kingma & Ba, 2017) with a learning rate of 0.001.

At each cycle, we select a fixed number of new samples and acquire the labels corresponding to them. Each experiment starts with the selection of one sample randomly and continues with the selection of the remaining samples with an acquisition function through active learning cycles. For each combination, we repeat the experiments eight times with different seeds.

**Hardware.** We utilized two different computing resources during the experiments. For the experiments with smaller models and datasets like Iris, we used an Intel(R) Core(TM) i9-10900KF processor paired with RTX 3090 GPU. For larger datasets and more complex models, which constitute the rest, we used an internal cluster with an Nvidia Tesla K80 GPU.

**Evaluation metrics.** We report the test accuracy on separate test sets for each dataset where the test sets for Mnist and FMnist consist of 384 samples while Iris has 30 samples, SVHN has 510 samples, Reuters has 450 samples, Emnist has 336 samples, Cifar-10 has 386 samples, and Tiny ImageNet has 296 samples. Additionally, we have separate validation sets of similar size to the test sets of each dataset.

**Time complexity.** The time complexity analysis consists of two parts: sample acquisition and model parameter estimation. Sample acquisition is identical for IMLE and DMLE, as they differ only in the model parameter estimation process. The complexity of the sample selection process is $\mathcal{O}(N_U(k + c))$ for both Top-$k$ and Stochastic Batch Selection methods, where $N_U$ is the number of samples in the unlabeled pool, $k$ is the size of the batch of samples selected in each cycle, and $c$ is the cost of making an acquisition call per sample.

Table 2: Comparison of mean classification test accuracies ±1 standard deviation for different sample selection sizes ($k$) during cycles and different sample selection strategies where number of samples in the labeled set $N_L = 100$. We use entropy for the acquisition function and SSMS (Stochastic Softmax Sampling), SPS (Stochastic Power Sampling), SSRS (Stochastic Soft-rank Sampling), and Top-$k$ sampling for the sample selection strategy. The bold text highlights the higher mean accuracy and lower standard deviation when comparing DMLE and IMLE under various sampling schemes.

| Dataset | $k$ | SSMS | | SPS | | SSRS | | Top-$k$ | |
|---|---|---|---|---|---|---|---|---|---|
| | | DMLE | IMLE | DMLE | IMLE | DMLE | IMLE | DMLE | IMLE |
| MNIST | 1 | **0.81 ± 0.03** | 0.79 ± 0.02 | **0.83 ± 0.03** | 0.82 ± 0.01 | **0.78 ± 0.04** | 0.75 ± 0.06 | **0.72 ± 0.07** | 0.69 ± 0.04 |
| | 5 | **0.82 ± 0.02** | 0.75 ± 0.03 | **0.82 ± 0.02** | 0.79 ± 0.03 | **0.75 ± 0.04** | 0.74 ± 0.04 | **0.68 ± 0.05** | 0.61 ± 0.07 |
| | 10 | **0.80 ± 0.02** | 0.76 ± 0.03 | **0.79 ± 0.03** | 0.77 ± 0.05 | **0.73 ± 0.05** | 0.71 ± 0.07 | **0.64 ± 0.08** | 0.48 ± 0.06 |
| EMNIST | 1 | **0.39 ± 0.03** | 0.37 ± 0.02 | **0.39 ± 0.02** | 0.36 ± 0.03 | **0.35 ± 0.04** | 0.33 ± 0.03 | **0.32 ± 0.03** | 0.30 ± 0.03 |
| | 5 | **0.39 ± 0.03** | 0.35 ± 0.02 | **0.40 ± 0.04** | 0.36 ± 0.01 | **0.36 ± 0.03** | 0.33 ± 0.02 | 0.31 ± 0.04 | **0.31 ± 0.03** |
| | 10 | **0.40 ± 0.01** | 0.37 ± 0.02 | **0.40 ± 0.02** | 0.38 ± 0.03 | **0.35 ± 0.02** | 0.34 ± 0.04 | **0.28 ± 0.03** | 0.24 ± 0.03 |
| REUTERS | 1 | **0.55 ± 0.02** | 0.47 ± 0.02 | **0.54 ± 0.01** | 0.50 ± 0.02 | **0.48 ± 0.05** | 0.46 ± 0.04 | **0.45 ± 0.06** | 0.40 ± 0.06 |
| | 5 | **0.54 ± 0.02** | 0.51 ± 0.02 | **0.52 ± 0.02** | 0.51 ± 0.03 | **0.47 ± 0.07** | 0.46 ± 0.04 | **0.47 ± 0.04** | 0.34 ± 0.06 |
| | 10 | **0.52 ± 0.01** | 0.49 ± 0.02 | **0.50 ± 0.02** | 0.48 ± 0.02 | **0.47 ± 0.04** | 0.46 ± 0.03 | **0.41 ± 0.04** | 0.32 ± 0.11 |
| SVHN | 1 | **0.30 ± 0.02** | 0.27 ± 0.03 | **0.30 ± 0.02** | 0.28 ± 0.03 | **0.25 ± 0.03** | 0.23 ± 0.03 | **0.22 ± 0.01** | 0.17 ± 0.03 |
| | 5 | **0.27 ± 0.05** | 0.23 ± 0.02 | **0.25 ± 0.04** | 0.24 ± 0.03 | **0.22 ± 0.03** | 0.20 ± 0.04 | **0.22 ± 0.02** | 0.19 ± 0.02 |
| | 10 | **0.26 ± 0.03** | 0.22 ± 0.01 | **0.24 ± 0.04** | 0.22 ± 0.01 | **0.21 ± 0.03** | **0.21 ± 0.03** | **0.21 ± 0.01** | 0.20 ± 0.03 |
| FMNIST | 1 | **0.74 ± 0.02** | 0.72 ± 0.03 | **0.75 ± 0.02** | 0.71 ± 0.02 | 0.70 ± 0.04 | **0.71 ± 0.04** | 0.52 ± 0.07 | **0.53 ± 0.06** |
| | 5 | **0.74 ± 0.02** | 0.71 ± 0.02 | **0.74 ± 0.02** | 0.73 ± 0.02 | **0.68 ± 0.06** | 0.66 ± 0.03 | **0.49 ± 0.08** | **0.49 ± 0.08** |
| | 10 | **0.73 ± 0.03** | 0.72 ± 0.02 | **0.72 ± 0.01** | 0.72 ± 0.04 | 0.69 ± 0.04 | **0.70 ± 0.04** | **0.48 ± 0.11** | 0.43 ± 0.05 |
| CIFAR-10 | 1 | **0.24 ± 0.02** | 0.21 ± 0.02 | **0.23 ± 0.02** | 0.22 ± 0.02 | **0.22 ± 0.03** | 0.21 ± 0.02 | 0.18 ± 0.03 | **0.19 ± 0.02** |
| | 5 | **0.22 ± 0.03** | 0.19 ± 0.01 | **0.22 ± 0.03** | 0.19 ± 0.02 | **0.21 ± 0.03** | 0.19 ± 0.03 | **0.17 ± 0.03** | **0.17 ± 0.03** |
| | 10 | **0.20 ± 0.04** | 0.17 ± 0.02 | **0.20 ± 0.02** | 0.16 ± 0.02 | **0.19 ± 0.02** | **0.19 ± 0.02** | **0.17 ± 0.03** | 0.15 ± 0.02 |
| IRIS | 1 | 0.96 ± 0.02 | **0.97 ± 0.01** | 0.96 ± 0.02 | **0.96 ± 0.01** | **0.97 ± 0.01** | **0.97 ± 0.01** | **0.99 ± 0.01** | 0.94 ± 0.02 |
| | 5 | **0.96 ± 0.03** | 0.87 ± 0.02 | **0.96 ± 0.03** | 0.87 ± 0.02 | **0.93 ± 0.03** | 0.86 ± 0.04 | **0.92 ± 0.05** | 0.81 ± 0.06 |
| | 10 | **0.93 ± 0.03** | 0.79 ± 0.03 | **0.93 ± 0.03** | 0.79 ± 0.03 | **0.85 ± 0.03** | 0.81 ± 0.02 | **0.85 ± 0.08** | 0.77 ± 0.05 |

The complexity of model parameter estimation differs between IMLE and DMLE due to the dependency term in the DMLE objective function, which requires additional calculations per sample. We perform the complexity analysis per sample in an epoch, assuming the use of Stochastic Gradient Descent for gradient updates and disregarding its associated complexity. Under these conditions, the time complexity of IMLE is $\mathcal{O}(N_L)$, while that of DMLE is $\mathcal{O}(cN_L)$, where $N_L$ represents the number of labeled samples. Experimental timing results are provided in Appendix E.

## 5.2 Results

**Accuracy results:** With the experiments, we seek to highlight the impact of not eliminating the dependency term during the model parameter estimation. Thus, the experiments present the test performance discrepancy between using DMLE vs. IMLE for multiple benchmark datasets with various sample selection schemes and acquisition functions. We provide the experiment results with the entropy and Core-set acquisition functions in this section. A similar case study for BALD and least confidence acquisition functions is presented in Appendix F.

Table 2 demonstrates the test accuracy results of using DMLE vs. IMLE after 100 collected samples across seven datasets, employing entropy as the acquisition function, with four different sample selection schemes—Stochastic Softmax Sampling (SSMS), Stochastic Power Sampling (SPS), Stochastic Soft-rank Sampling (SSRS), and Top-$k$ sampling—along with batch sizes $k \in \{1, 5, 10\}$. When SSMS or SPS are used for the sample selection, DMLE leads to better test accuracy results for all datasets and batch sizes except for the Iris dataset with $k = 1$. In the case of SSRS as the sample selection strategy, test accuracies are either improved when DMLE is employed as the method of model parameter estimation or competitive with IMLE, except for FMnist with $k = 1$ or $k = 10$. For Top-$k$ selection, DMLE either outperforms or leads to

comparable results with IMLE in all cases except Emnist with $k = 5$, FMnist with $k = 1$, and Cifar-10 with $k = 1$. The average test accuracy improvements of 9.6%, 7%, 3.9%, and 11.8% are observed for SSMS, SPS, SSRS, and Top-$k$ sampling strategies respectively, which underline the significance of sample dependencies in model parameter estimation. Using different $k$ values for the sample selection size yields average test accuracy improvements of 6%, 8.6%, and 10.5% for $k = 1$, $k = 5$, and $k = 10$ respectively. We observe performance degradation as the sample selection size $k$ increases for both DMLE and IMLE. We conjecture that this drop occurs due to the reduced diversity in the collected samples, where similar samples might have been collected in an active learning cycle. Even though we observe a trend for the test accuracy to drop when $k$ is increased, the reported average accuracy improvements for different $k$ values suggest that DMLE leads to better results than IMLE. The experiments in Appendix F) further underline the importance of using DMLE in active learning for model parameter estimation and its impact on model performance.

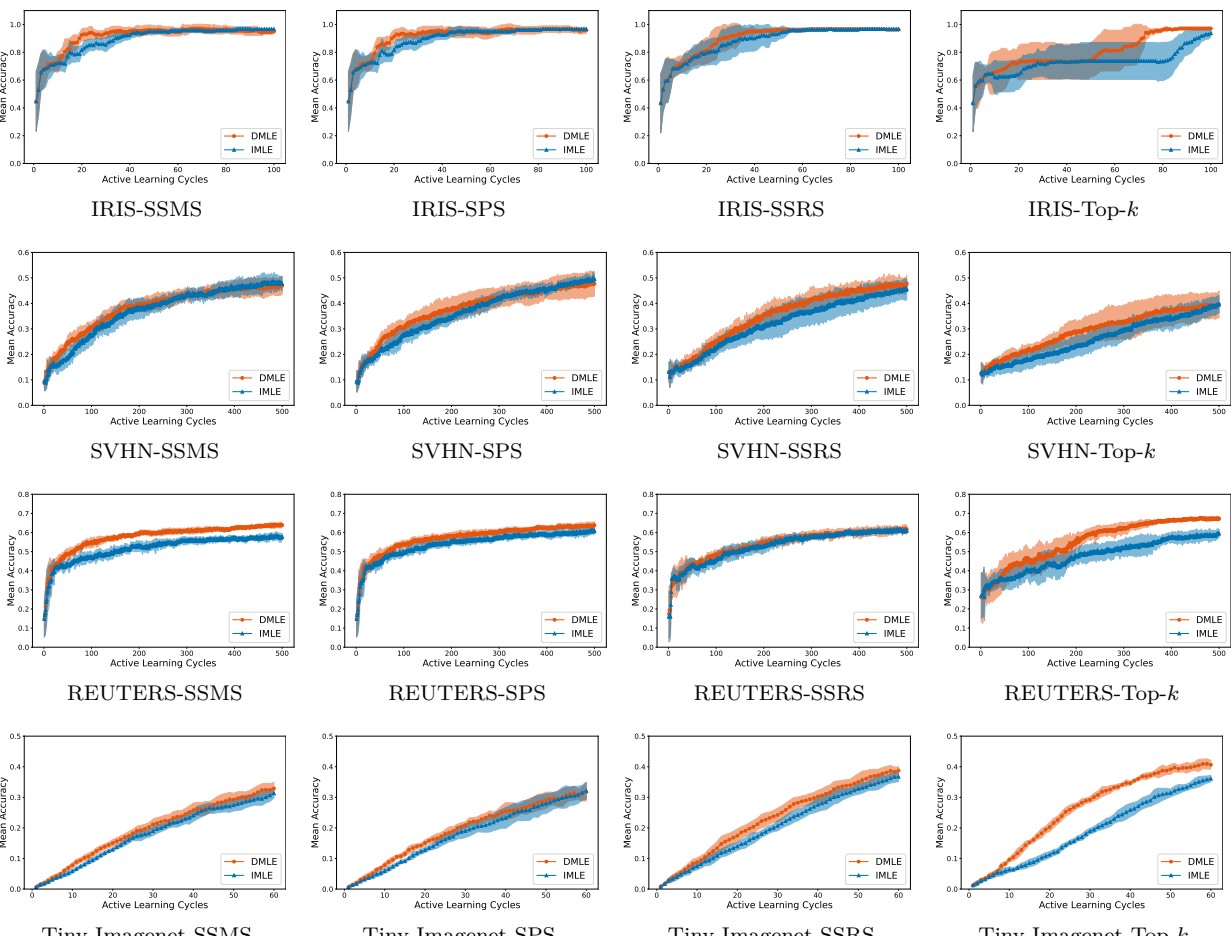

Figure 3: The average test accuracy comparison with ±1 standard deviation for DMLE and IMLE over cycles for Iris, SVHN, Reuters, and Tiny ImageNet datasets for different sample selection strategies, namely Stochastic Softmax Sampling(SSMS), Stochastic Power Sampling(SPS), Stochastic Soft-rank Sampling(SSRS), and Top-$k$ Sampling where sample selection size $k = 1$ for all except Tiny ImageNet and $k = 5$ for Tiny ImageNet.

To analyze the accuracy evolution over the active learning cycles we plot the mean test accuracy ±1 standard deviation for DMLE and IMLE using four sample selection strategies with entropy acquisition function. Figure 3 shows such curves for the Iris, SVHN, Reuters, and Tiny ImageNet datasets where the sample selection size is $k = 1$ for the first three and $k = 5$ for Tiny ImageNet, with the remaining four datasets (MNIST, Fashion-Mnist, Emnist, and Cifar-10) provided in Appendix F. In the first row of Figure 3, the test

accuracy performance for the Iris dataset after 100 cycles of collecting one sample at each cycle is shown for different sample selection schemes. The initial unlabeled set size for the Iris dataset is 110 which means after 100 cycles, we collected almost all of the samples in the unlabeled set. Thus, it is observed that DMLE and IMLE exhibit similar performance in the later cycles, while DMLE yields better test accuracy in the earlier cycles, demonstrating its advantage within the active learning framework. In the second and third rows of Figure 3, the average test accuracy plots for 500 cycles on SVHN and Reuters datasets are given. The improvement gained by accounting for sample dependencies during model parameter estimation with DMLE is evident in all plots, regardless of the sample selection strategy employed. In the last row of Figure 3, the performance difference between DMLE and IMLE is observable across all selection strategies for the Tiny ImageNet dataset, with DMLE showing a particularly greater improvement when Top-$k$ selection is used.

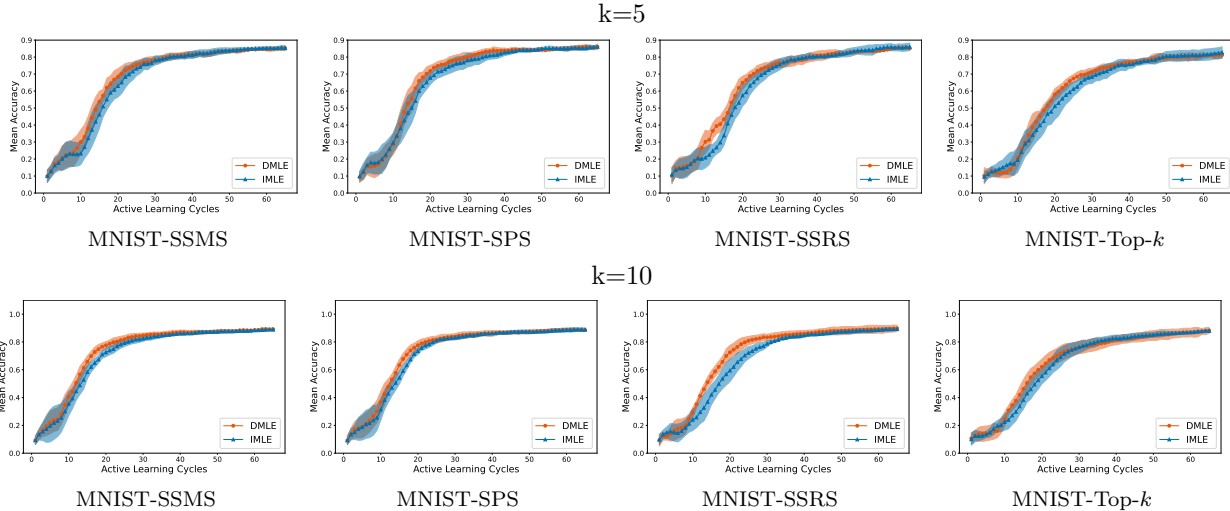

Figure 4: The experiments comparing DMLE and IMLE using the clustering-based Coreset approach with different sample selection strategies—SSMS, SPS, SSRS, and Top-$k$ Sampling—are presented. The average test accuracy plots with $\pm 1$ standard deviation over cycles are shown, with sample selection size $k = 5$ for the first row and $k = 10$ for the second row. The plots demonstrate that DMLE outperforms IMLE, particularly in the earlier cycles, consistent with our observations using uncertainty-based acquisition functions.

In Figure 4, we provide the accuracy plots for the Core-set acquisition score, which is a diversity-based clustering approach for sample selection where $k = 5, 10$. It can be observed that, across all sample selection strategies combined with this acquisition function, DMLE consistently outperforms IMLE from the earlier cycles onward. This demonstrates that taking into account the dependency term during model parameter estimation is an important problem regardless of the acquisition function and can be applicable to any active learning setup as long as acquisition scores/ranks can be obtained.

**Hypothesis testing:** We used the Wilcoxon signed-rank test (Conover, 1971) to compare the performance of DMLE and IMLE, based on 8 random seeds per experiment for the experiments shown in Table 2, Table 4, and Table 5. As this non-parametric test does not assume normality, it is suitable for cases where the differences between paired observations may not follow a normal distribution, such as accuracy values in active learning experiments with varying random initializations. Our results show that DMLE outperforms IMLE in 92% of these experiments, with statistical significance ($p < 0.05$). This suggests that DMLE outperforms IMLE significantly, providing strong evidence of its effectiveness in improving model accuracy by accounting for sample dependencies.

## 6    Conclusion and Future Works

In this paper, we introduced Dependency-aware Maximum Likelihood Estimation (DMLE) for the active learning framework, a novel approach that highlights the incompatibility of the i.i.d. data assumption in

conventional parameter estimation methods with the active learning paradigm. Specifically, DMLE preserves the sample dependency that would otherwise be overlooked in conventional independent MLE (IMLE), ensuring they are properly taken into account in the estimation. We examined the impact of this dependency term across various acquisition functions, sample selection strategies, and sample sizes, evaluating its effect on multiple benchmark datasets with varying model complexities. Our experiments revealed a significant improvement in average test accuracy when using DMLE compared to IMLE, underlining the critical role of accounting for sample dependencies in model parameter estimation.

A promising direction for future work is the theoretical characterization of the dependency term's distribution under Top-$k$ selection strategies, beyond the current empirical analysis we provide. This would lead to deeper insights into the approximation behavior of DMLE and help identify scenarios where its benefits may diminish. Additionally, examining DMLE's performance across varying uncertainty and diversity conditions represents a valuable line of research, with potential to inform improvements in the acquisition process. Another promising direction is exploring connections between DMLE and submodular optimization frameworks for large-batch selection. Prior work has shown that submodularity can effectively guide the selection of informative and diverse subsets under efficiency constraints (Golovin & Krause, 2017). Adapting DMLE within such a combinatorial framework may help scale its benefits to larger selection sizes while maintaining tractable optimization.

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

## Appendix

## A  Stochastic Batch Selection

Recently, Kirsch et al. (2021) have proposed Stochastic Batch Acquisition as a probabilistic sampling strategy in active learning, which is known to outperform conventional Top-$k$ sampling while avoiding additional computational expenses. Their approach includes three different stochastic batch acquisition methods depending on the perturbation method they use: Softmax, Power, and Soft-rank acquisition where each corresponds to sampling from the corresponding probability distribution given in Table 1. We generate the sample set $S_{t+1}$ by assuming we sample from a probability distribution without replacement when we make a batch selection. Under this assumption, the probability that we sample the sequence of elements $\mathbf{s}_{t+1} = [x_1, x_2, \ldots, x_k]$ in this order depends on the stochastic sampling method and defined as follows:

- **Softmax acquisition**

$$P(\mathbf{s}_{t+1} \mid D_t; \theta) = \frac{1}{Z_t} \prod_{i=1}^{k} e^{\beta a(x_i, D_t, \theta)} \quad \text{where} \quad Z_{t,i} \equiv \sum_{x \in U_t \setminus \{x_1, \ldots, x_{i-1}\}} e^{\beta a(x, D_t, \theta)}.$$

- **Power acquisition**

$$P(\mathbf{s}_{t+1} \mid D_t; \theta) = \frac{1}{Z_t} \prod_{i=1}^{k} a(x_i, D_t, \theta)^{\beta} \quad \text{where} \quad Z_{t,i} \equiv \sum_{x \in U_t \setminus \{x_1, \ldots, x_{i-1}\}} a(x, D_t, \theta)^{\beta}.$$

- **Soft-rank acquisition**

$$P(\mathbf{s}_{t+1} \mid D_t; \theta) = \frac{1}{Z_t} \prod_{i=1}^{k} r_i^{-\beta} \quad \text{where} \quad Z_{t,i} \equiv \sum_{x \in U_t \setminus \{x_1, \ldots, x_{i-1}\}} r^{-\beta}$$

where for all acquisition methods the normalization constant $Z_t \equiv \prod_{i=1}^{k} Z_{t,i}$, $r$ is the descending ranking of the acquisition scores $a(x, D_t, \theta)$, with the smallest rank being 1, and $\beta > 0$ is a temperature parameter: the higher it is, the closer the selection is to Top-$k$.

Note that this probability characterizes the sequence of elements we sample, so we use $\mathbf{s}_{t+1}$ to distinguish it from (unordered) set $S_{t+1}$. We also denote by

$$\mathbf{d}_{t+1} = [(x_1, y_1), (x_2, y_2), \ldots, (x_k, y_k)]$$

the corresponding sequence of collected tuples of data points and labels. Moreover

$$P(\mathbf{d}_{t+1} \mid D_t; \theta) = P(\mathbf{s}_{t+1} \mid D_t; \theta) \prod_{i=1}^{k} P(y_i \mid x_i; \theta), \tag{12}$$

where $D_0 \equiv \emptyset$.

We couple our method DMLE with these sample selection procedures. We explicitly model sample dependency and implement it via sampling from the probability distributions given in Table 1 to model $P(S_\tau \mid D_{\tau-1}; \theta)$ term which leads to the objective functions for the following stochastic sampling strategies:

- **Softmax acquisition**

$$\hat{\theta}_{t+1}^{DMLE} = \arg\max_{\theta} \sum_{\tau=1}^{t+1} \sum_{i=1}^{k} \left[ \ln P(y_{\tau,i} \mid x_{\tau,i}; \theta) + \ln \frac{e^{\beta a(x_{\tau,i}, D_{\tau-1}, \theta)}}{Z_{\tau,i}} \right] \tag{13}$$

$$= \arg\max_{\theta} \sum_{(x,y) \in D_{t+1}} \ln P(y \mid x; \theta) + \sum_{\tau=1}^{t+1} \sum_{i=1}^{k} \ln \frac{e^{\beta a(x_{\tau,i}, D_{\tau-1}, \theta)}}{Z_{\tau,i}} \tag{14}$$

- **Power acquisition**

$$\hat{\theta}_{t+1}^{DMLE} = \arg\max_\theta \sum_{\tau=1}^{t+1} \sum_{i=1}^{k} \left[ \ln P(y_{\tau,i} \mid x_{\tau,i}; \theta) + \ln \frac{a(x_{\tau,i}, D_{\tau-1}, \theta)^\beta}{Z_{\tau,i}} \right] \tag{15}$$

$$= \arg\max_\theta \sum_{(x,y) \in D_{t+1}} \ln P(y \mid x; \theta) + \sum_{\tau=1}^{t+1} \sum_{i=1}^{k} \ln \frac{a(x_{\tau,i}, D_{\tau-1}, \theta)^\beta}{Z_{\tau,i}} \tag{16}$$

- **Soft-rank acquisition**

$$\hat{\theta}_{t+1}^{DMLE} = \arg\max_\theta \sum_{\tau=1}^{t+1} \sum_{i=1}^{k} \left[ \ln P(y_{\tau,i} \mid x_{\tau,i}; \theta) + \ln \frac{r_{\tau,i}^{-\beta}}{Z_{\tau,i}} \right] \tag{17}$$

$$= \arg\max_\theta \sum_{(x,y) \in D_{t+1}} \ln P(y \mid x; \theta) + \sum_{\tau=1}^{t+1} \sum_{i=1}^{k} \ln \frac{r_{\tau,i}^{-\beta}}{Z_{\tau,i}} \tag{18}$$

Thus, depending on the sampling strategy we follow in active learning, the dependency term that appears in the model parameter estimation objective varies. For the active learning framework that uses the Stochastic Softmax acquisition function, equation 14 is utilized for the model parameter estimation objective, while equation 16 is used for the Stochastic Power acquisition function, and equation 18 for the Stochastic Soft-rank acquisition function.

**Evaluation of the normalization constant:** To analyze the relevance of the $Z_{\tau,i}$ term in equation 14, equation 16 and equation 18, we experimented with the MNIST dataset where the trend for the term over the active learning cycles is observed. Figure 5 shows the evolution of this term for different sample selection strategies. Notably, the resulting curves are nearly constant for all. These results suggest a simplified optimization problem where the normalization constant is neglected, which is followed for the experiments in this manuscript.

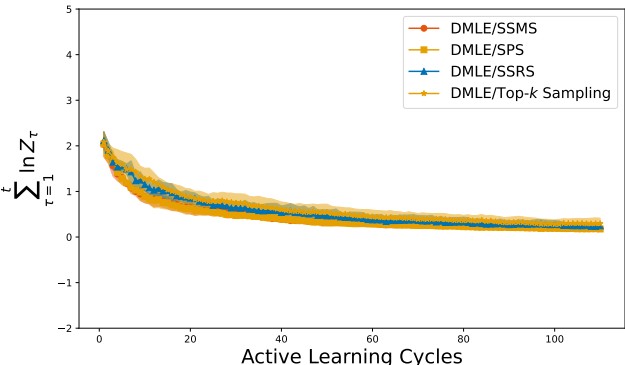

Figure 5: The change of the term $\sum_{\tau=1}^{t} ln(Z_\tau)$ in the model parameter estimation objective function through the active learning cycles for the MNIST dataset. One can note that the term changes marginally over the cycles which motivates the elimination of this term from the model parameter estimation while taking into account the computational expenses it introduces into the process.

## B    Independent Maximum Likelihood Estimation (IMLE) Derivation in Active Learning

Due to the sequential collection of the labeled training data, we can write the log-likelihood formula with the cycle notation as follows:

$$\hat{\theta}_{t+1}^{\mathrm{MLE}} = \arg\max_{\theta} \sum_{\tau=1}^{t+1} \ln P(\partial D_\tau \mid D_{\tau-1}; \theta) \tag{19}$$

$$= \arg\max_{\theta} \sum_{\tau=1}^{t+1} \ln P(x_{\tau,k}, y_{\tau,k}, \dots, x_{\tau,1}, y_{\tau,1} \mid D_{\tau-1}; \theta) \tag{20}$$

With the application of product rule:

$$\hat{\theta}_{t+1}^{\mathrm{MLE}} = \arg\max_{\theta} \sum_{\tau=1}^{t+1} \left[ \ln P(y_{\tau,k}, \dots, y_{\tau,1} \mid x_{\tau,k}, \dots, x_{\tau,1}, D_{\tau-1}; \theta) + \ln P(x_{\tau,k}, \dots, x_{\tau,1} \mid D_{\tau-1}; \theta) \right] \tag{21}$$

By relying on the modeling assumption of $f_\theta : x \to y$:

$$\hat{\theta}_{t+1}^{\mathrm{MLE}} = \arg\max_{\theta} \sum_{\tau=1}^{t+1} \left[ \sum_{i=1}^{k} \ln P(y_{\tau,i} \mid x_{\tau,i}; \theta) + \ln P(x_{\tau,k}, \dots, x_{\tau,1} \mid D_{\tau-1}; \theta) \right] \tag{22}$$

As $S_\tau$ is the samples selected in a cycle for labeling, $\{x_{\tau,k}, \dots, x_{\tau,1}\}$ can be written as follows:

$$\hat{\theta}_{t+1}^{\mathrm{MLE}} = \arg\max_{\theta} \sum_{\tau=1}^{t+1} \left[ \sum_{i=1}^{k} \ln P(y_{\tau,i} \mid x_{\tau,i}; \theta) + \ln P(S_\tau \mid D_{\tau-1}; \theta) \right] \tag{23}$$

The i.i.d. sample assumption in conventional Maximum Likelihood Estimation causes $\ln P(S_\tau \mid D_{\tau-1}; \theta)$ term to be a constant during the estimation which leads to the elimination of that term as follows:

$$\hat{\theta}_{t+1}^{\mathrm{MLE}} = \arg\max_{\theta} \sum_{\tau=1}^{t+1} \left[ \sum_{i=1}^{k} \ln P(y_{\tau,i} \mid x_{\tau,i}; \theta) \right] \tag{24}$$

$$= \arg\max_{\theta} \sum_{(x,y) \in D_{t+1}} \ln P(y \mid x; \theta). \tag{25}$$

## C  Proofs

### C.1  Proof of Theorem 1.

The Fisher Information matrix quantifies the precision of parameter estimates, where variance is inversely related to Fisher Information. For simplicity in the proof, we consider a sample selection size of $k = 1$ at each cycle, though the result can be extended to larger $k$.

For IMLE:

$$L^{\mathrm{IMLE}}(\theta) = \sum_{\tau=1}^{t} \ln P(y_\tau \mid x_\tau; \theta), \tag{26}$$

where $y_\tau$ are labels, and $x_\tau$ are inputs, and $\theta$ is the parameters.

For DMLE:

$$L^{\mathrm{DMLE}}(\theta) = \sum_{\tau=1}^{t} \left[ \ln P(y_\tau \mid x_\tau; \theta) + \ln P(x_\tau \mid D_{\tau-1}; \theta) \right], \tag{27}$$

where $P(x_\tau \mid D_{\tau-1}; \theta)$ accounts for sample dependency.

The individual Fisher Information terms are given by:

$$I_\tau^{(y)}(\theta) = \mathbb{E}\left[-\frac{\partial^2}{\partial\theta\partial\theta^T}\ln P(y_\tau \mid x_\tau; \theta)\right], \tag{28}$$

$$I_\tau^{(x)}(\theta) = \mathbb{E}\left[-\frac{\partial^2}{\partial\theta\partial\theta^T}\ln P(x_\tau \mid D_{\tau-1}; \theta)\right]. \tag{29}$$

The Fisher Information matrices for IMLE and DMLE are:

$$I(\theta^{\text{IMLE}}) = \sum_{\tau=1}^{t} I_\tau^{(y)}(\theta), \tag{30}$$

$$I(\theta^{\text{DMLE}}) = \sum_{\tau=1}^{t}\left[I_\tau^{(y)}(\theta) + I_\tau^{(x)}(\theta)\right]. \tag{31}$$

Matching corresponding terms, we see:

$$I(\theta^{\text{DMLE}}) = I(\theta^{\text{IMLE}}) + \sum_{\tau=1}^{t} I_\tau^{(x)}(\theta). \tag{32}$$

Since $I_\tau^{(x)}(\theta)$ is positive semi-definite:

$$I(\theta^{\text{DMLE}}) \succeq I(\theta^{\text{IMLE}}). \tag{33}$$

The Cramér-Rao Lower Bound (CRLB) indicates:

$$\text{Cov}(\hat{\theta}) \succeq I(\theta)^{-1}. \tag{34}$$

Since $I(\theta^{\text{DMLE}}) \succeq I(\theta^{\text{IMLE}})$:

$$\text{Cov}(\hat{\theta}^{\text{DMLE}}) \preceq \text{Cov}(\hat{\theta}^{\text{IMLE}}). \tag{35}$$

Thus, DMLE yields lower variance guarantees and more precise estimates compared to IMLE. ∎

## C.2 Proof of Theorem 2.

The expected log-likelihood of the learned model $\hat{\theta}$ over the true data distribution $P^{true}$ is given by:

$$\mathbb{E}_{(x',y')\sim P^{true}}\left[\ln L(\hat{\theta})\right] = \mathbb{E}_{(x',y')\sim P^{true}}\left[\ln P(y' \mid x'; \hat{\theta}) + \ln P(x' \mid D_t; \hat{\theta})\right]. \tag{36}$$

where $\hat{\theta} = \arg\max_\theta \ln P(D_{t+1}; \theta) = \arg\max_\theta \sum_{(x,y)\in D_{t+1}} \ln P(y \mid x; \theta) + \sum_{\tau=1}^{t+1}\sum_{x\in S_\tau} \ln P(x|D_{\tau-1}; \theta)$.
Expanding the expectation:

$$\sum_{x'} P(x') \sum_{y'} P(y' \mid x')\left[\ln P(y' \mid x'; \hat{\theta}) + \ln P(x' \mid D_t; \hat{\theta})\right]. \tag{37}$$

Rewriting as separate summations:

$$\sum_{x'} P(x') \sum_{y'} P(y' \mid x') \ln P(y' \mid x'; \hat{\theta}) + \sum_{x'} P(x') \ln P(x' \mid D_t; \hat{\theta}). \tag{38}$$

Using KL divergence and entropy identities:

$$-D_{\mathrm{KL}}(P(y' \mid x') \parallel P(y' \mid x'; \hat{\theta})) + H(P(y' \mid x')) - D_{\mathrm{KL}}(P(x') \parallel P(x' \mid D_t; \hat{\theta})) + H(P(x')). \tag{39}$$

Since $H(P(y' \mid x'))$ and $H(P(x'))$ are constants, they can be omitted:

$$\mathbb{E}_{(x',y') \sim P^{true}}\left[\ln L(\hat{\theta})\right] = -D_{\mathrm{KL}}(P(y' \mid x') \parallel P(y' \mid x'; \hat{\theta})) - D_{\mathrm{KL}}(P(x') \parallel P(x' \mid D_t; \hat{\theta})). \tag{40}$$

This result shows that maximizing the expected log-likelihood is equivalent to minimizing the sum of two KL divergences, ensuring that both the predictive and data distributions of the model remain close to the true distributions. ∎

## D   Comparison with Farquhar et al. (2021)

We conducted experiments using the MNIST dataset with entropy as the acquisition function and various sample selection methods. For model parameter estimation, we utilized IMLE, DMLE, and the statistical bias mitigation approach proposed by Farquhar et al. (2021). The results are consistent with the findings of Farquhar et al. (2021), which show that mitigating bias with their proposed method leads to performance degradation with neural networks compared to conventional MLE (IMLE) in active learning. In contrast, our Dependency-aware MLE (DMLE) enhances model performance.

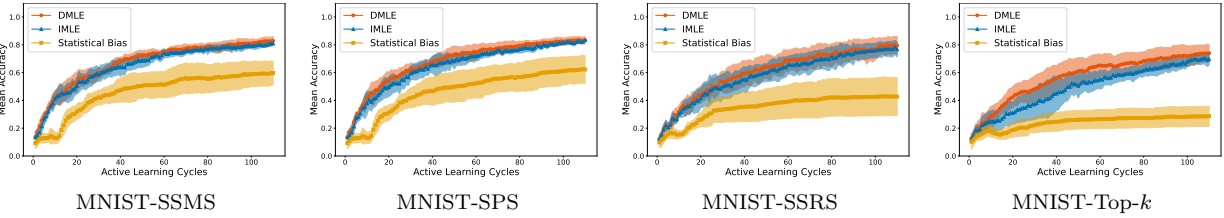

Figure 6: The experiments comparing DMLE and IMLE with statistical bias mitigation method proposed by Farquhar et al. (2021) where different sample selection strategies—SSMS, SPS, SSRS, and Top-$k$ Sampling—are employed. The average test accuracy plots with ±1 standard deviation over cycles are shown for sample selection size $k = 1$. The plots demonstrate that DMLE outperforms the bias mitigation method proposed by Farquhar et al. (2021).

## E   Experimental Timing Results

To demonstrate that DMLE does not impose a significant computational burden during training, we measured elapsed times using a timer. Table 3 presents the mean elapsed times for DMLE versus IMLE, combined with various sample selection functions and entropy as the acquisition function, with selection sizes of k=1, k=5, or k=10 on the MNIST dataset. We selected MNIST due to its large unlabeled pool of 30,000 samples. In the table, the additional time required to include dependency with DMLE is negligible compared to the overall elapsed times. Additionally, the mean extra time due to DMLE is 1.88 minutes across all combinations, indicating that the time complexity remains largely unaffected.

Table 3: Comparison of mean elapsed times for DMLE and IMLE with different sample selection sizes $k$ and number of cycles $T$ on the MNIST dataset (30000 samples in the unlabeled pool at the beginning). The table shows the mean elapsed times over 8 experiments per combination for $k = 1$, $k = 5$, and $k = 10$. On average, DMLE causes an additional 1.88 minutes of elapsed time compared to IMLE for Stochastic Softmax Sampling (SSMS), Stochastic Power Sampling (SPS), and Stochastic Soft-rank Sampling (SSRS).

| k | T | Sampling Strategy | DMLE Time | IMLE Time |
|---|---|---|---|---|
| 1 | 2500 | SSMS | 465:30.31 min | 462:50.19 min |
| | | SPS | 473:30.78 min | 471:19.08 min |
| | | SSRS | 477:17.79 min | 467:51.34 min |
| 5 | 500 | SSMS | 92:58.25 min | 92:28.56 min |
| | | SPS | 92:54.06 min | 91:12.24 min |
| | | SSRS | 92:28.00 min | 92:48.54 min |
| 10 | 250 | SSMS | 46:03.89 min | 45:57.16 min |
| | | SPS | 48:00.51 min | 48:52.36 min |
| | | SSRS | 49:03.53 min | 49:18.87 min |

As the difference between DMLE and IMLE lies in the objective function, we also plot the loss processing times per cycle to better observe the computational expense introduced by the dependency term in DMLE, as shown in Figure 7. This experiment, conducted on the MNIST dataset collecting 500 samples with sample selection size $k = 1$, shows that the loss processing time increases for both methods across cycles as more samples are labeled. Although the difference between DMLE and IMLE grows over time, it remains relatively small in practice.

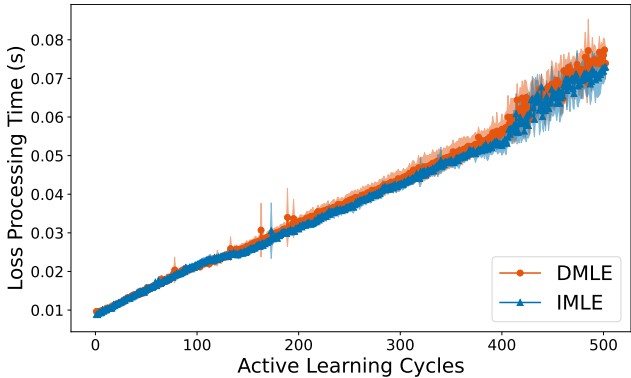

Figure 7: Per-cycle loss processing time for DMLE and IMLE during active learning on MNIST (500 cycles, sample selection size $k = 1$). This plot illustrates how the computational cost evolves across cycles, emphasizing the impact of DMLE's dependency-aware loss formulation. While both methods exhibit increased processing times as the labeled set grows, DMLE shows a steeper rise due to the additional overhead of modeling dependencies. Nonetheless, within the context of active learning, where the objective is to achieve high performance with a limited number of labeled samples, the overall runtime remains practical, demonstrating that the additional computational cost of DMLE is manageable in typical active learning scenarios.

## F    Additional Results

To further analyze the importance of taking the sample dependencies into account during model parameter estimation in active learning, we plot the test accuracy plots with $\pm 1$ standard deviation for four more datasets, e.g., Mnist, Fashion-Mnist (FMnist), Emnist, and Cifar-10, where for each we utilize four different sample selection strategies explained in Section 3.2. In Figure 8, we observe the performance improvement with DMLE when compared with IMLE in most cases. For the cases where DMLE doesn't outperform IMLE, we believe further hyperparameter tuning for $\beta$ might lead to better performance. For all experiments here, we use $\beta = 1$ as suggested by Kirsch et al. (2021).

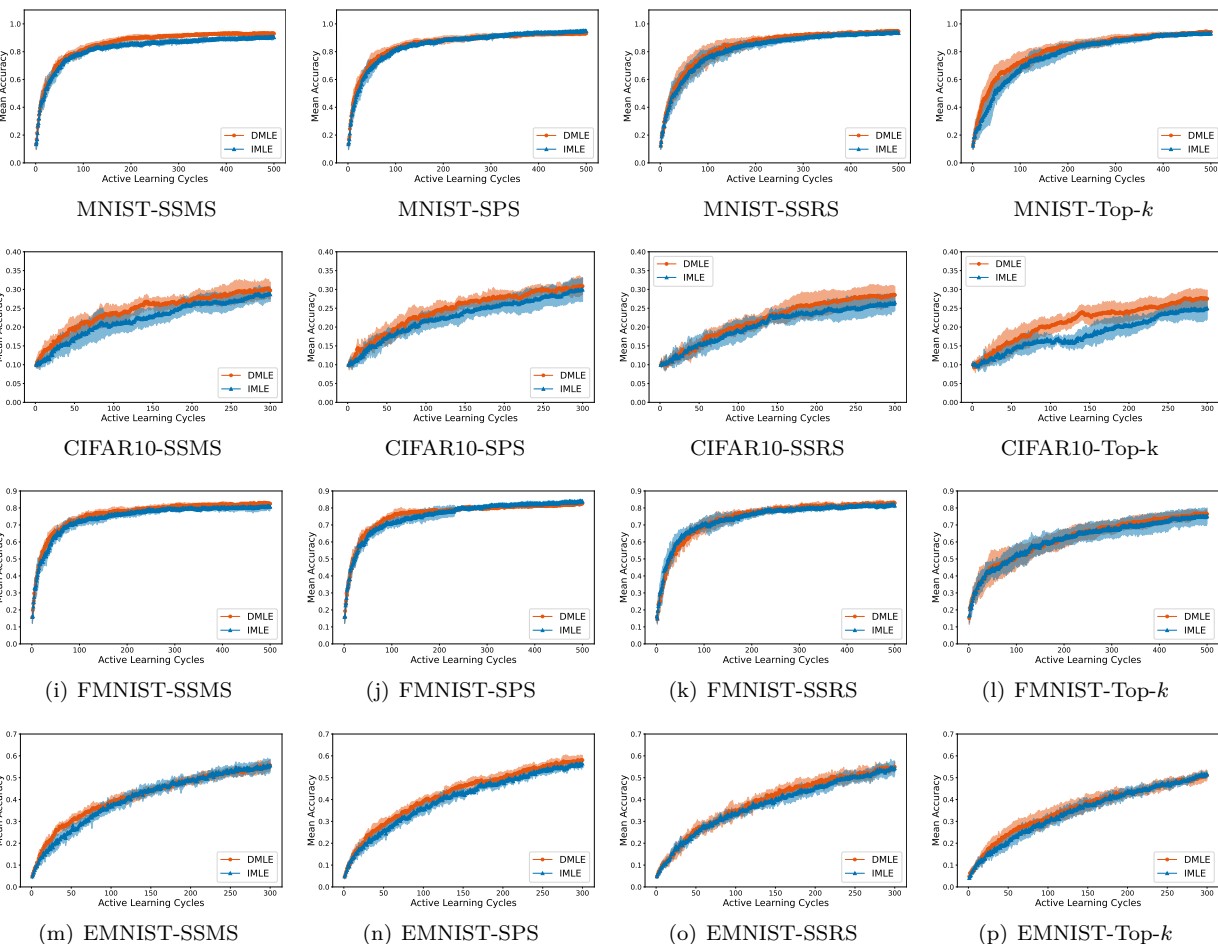

Figure 8: The average test accuracy comparison with $\pm 1$ standard deviation for DMLE and IMLE over cycles for Mnist, Cifar-10, FMnist, and Emnist datasets for different sample selection strategies, namely Stochastic Softmax Sampling(SSMS), Stochastic Power Sampling(SPS), Stochastic Soft-rank Sampling(SSRS), and Top-$k$ Sampling where sample selection size is $k = 1$. In the earlier cycles, DMLE was observed to outperform IMLE in all cases except for FMnist with SSRS and Top-k which is highly advantageous in active learning.

In addition to providing average test accuracy results in Table 2 for multiple datasets with different $k$ values used for sampling at each cycle until 100 samples are collected, we also plot the accuracy improvement over cycles for the Iris dataset to better observe the importance of DMLE. In Figure 9, we observe that when $k = 1$, the curves for DMLE and IMLE tend to converge to the same accuracy. However, with $k = 5$ and $k = 10$, using DMLE becomes significantly more important, as the resulting accuracy values for DMLE and IMLE diverge significantly.

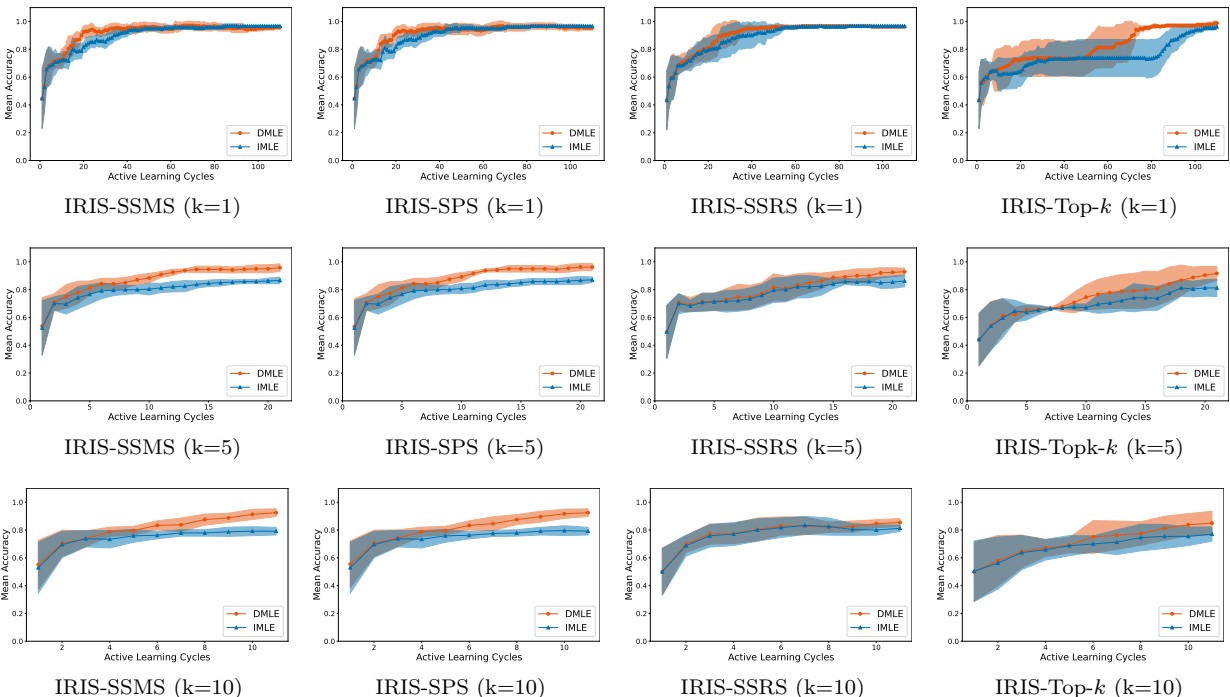

Figure 9: The average test accuracy comparison with ±1 standard deviation for DMLE and IMLE over cycles for Iris for different sample selection strategies, namely Stochastic Softmax Sampling(SSMS), Stochastic Power Sampling(SPS), Stochastic Soft-rank Sampling(SSRS), and Top-$k$ Sampling where sample selection sizes are $k = 1$, $k = 5$, and $k = 10$. As the sample selection size increases, the significance of using DMLE over IMLE becomes more acute.

Table 4 is based on the same experimental setup as Table 2 where the only difference is the acquisition function used for the experiments. For these results, we use the BALD acquisition function. The average test accuracy improvements resulting from using DMLE over IMLE for different sample selection schemes are 3.5%, 4.1%, 3.8%, and 5.5% for SSMS, SPS, SSRS, and Top-$k$ respectively. Moreover, for $k = 1$, $k = 5$, and $k = 10$, the average test accuracy improvements of 7.6%, 5.2%, and 3.8% are attained when DMLE is employed. Both with different sample selection strategies and sample selection sizes, it is notable that taking the sample dependency into account via DMLE during model parameter estimation elicits better average test accuracy performance when compared with IMLE.

In Table 5 we use the same experimental setup as Table 2 with Least Confidence as the acquisition function. The average test accuracy improvements obtained when DMLE used instead of IMLE for different sample selection strategies are 4.6%, 3.7%, 2.3%, and 3.8% for SSMS, SPS, SSRS, and Top-$k$ respectively. When we experiment with different sample selection sizes for sample acquisition at each cycle, we see average test accuracy improvements of 1.6%, 4.9%, and 5% for $k = 1$, $k = 5$, and $k = 10$ respectively. Both experiments with different sample selection strategies and sample selection sizes demonstrate the importance of accounting for sample dependencies during model parameter estimation in active learning.

# G   Impact Statement

The method we propose may impact fields where acquiring labeled data is scarce, dangerous, or costly by allowing better performance with less data. This is the case in many problems in the medical, bioinformatics, remote sensing, and materials fields, to name but a few.

Table 4: Comparison of mean classification test accuracies $\pm 1$ standard deviation for different sample selection sizes ($k$) during cycles and different sample selection strategies where number of samples in the labeled set $N_L = 100$. We use BALD for the acquisition function and SSMS (Stochastic Softmax Sampling), SPS (Stochastic Power Sampling), SSRS (Stochastic Soft-rank Sampling), and Top-$k$ sampling for the sample selection strategy. The bold text highlights the higher mean accuracy and lower standard deviation when comparing DMLE and IMLE under various sampling schemes.

| Dataset | $k$ | SSMS | | SPS | | SSRS | | Top-$k$ | |
|---|---|---|---|---|---|---|---|---|---|
| | | DMLE | IMLE | DMLE | IMLE | DMLE | IMLE | DMLE | IMLE |
| MNIST | 1 | **0.77 ± 0.02** | 0.77 ± 0.03 | 0.77 ± 0.03 | **0.78 ± 0.03** | **0.82 ± 0.02** | 0.80 ± 0.03 | 0.80 ± 0.05 | **0.80 ± 0.04** |
| | 5 | **0.76 ± 0.01** | 0.75 ± 0.02 | **0.76 ± 0.03** | 0.73 ± 0.03 | **0.79 ± 0.03** | 0.78 ± 0.02 | **0.80 ± 0.03** | 0.79 ± 0.02 |
| | 10 | **0.76 ± 0.03** | **0.76 ± 0.03** | **0.76 ± 0.04** | 0.75 ± 0.02 | **0.79 ± 0.03** | 0.76 ± 0.02 | **0.79 ± 0.03** | 0.78 ± 0.02 |
| EMNIST | 1 | **0.36 ± 0.02** | 0.36 ± 0.03 | **0.37 ± 0.02** | 0.35 ± 0.04 | **0.36 ± 0.03** | **0.36 ± 0.03** | **0.36 ± 0.03** | 0.34 ± 0.01 |
| | 5 | **0.38 ± 0.02** | **0.38 ± 0.02** | 0.36 ± 0.02 | **0.37 ± 0.02** | **0.35 ± 0.02** | **0.35 ± 0.02** | 0.36 ± 0.03 | **0.37 ± 0.04** |
| | 10 | **0.36 ± 0.03** | 0.36 ± 0.04 | **0.37 ± 0.03** | 0.35 ± 0.03 | 0.36 ± 0.03 | **0.38 ± 0.02** | **0.38 ± 0.03** | 0.36 ± 0.04 |
| REUTERS | 1 | **0.51 ± 0.02** | 0.49 ± 0.02 | 0.48 ± 0.03 | **0.48 ± 0.02** | **0.51 ± 0.02** | 0.50 ± 0.03 | **0.50 ± 0.01** | 0.48 ± 0.04 |
| | 5 | **0.51 ± 0.03** | **0.51 ± 0.03** | **0.51 ± 0.01** | **0.51 ± 0.01** | **0.49 ± 0.03** | 0.48 ± 0.01 | **0.48 ± 0.01** | 0.47 ± 0.01 |
| | 10 | **0.48 ± 0.04** | 0.47 ± 0.04 | **0.48 ± 0.02** | 0.47 ± 0.01 | **0.46 ± 0.01** | 0.45 ± 0.03 | **0.44 ± 0.01** | 0.42 ± 0.04 |
| SVHN | 1 | **0.28 ± 0.03** | 0.26 ± 0.03 | **0.31 ± 0.03** | 0.25 ± 0.02 | **0.30 ± 0.02** | 0.28 ± 0.05 | **0.26 ± 0.04** | 0.22 ± 0.04 |
| | 5 | **0.24 ± 0.02** | 0.23 ± 0.03 | **0.27 ± 0.04** | 0.23 ± 0.02 | **0.25 ± 0.03** | **0.25 ± 0.03** | 0.23 ± 0.04 | **0.23 ± 0.02** |
| | 10 | **0.22 ± 0.03** | **0.22 ± 0.03** | **0.24 ± 0.03** | 0.23 ± 0.03 | 0.23 ± 0.05 | **0.23 ± 0.04** | **0.22 ± 0.03** | 0.21 ± 0.03 |
| FMNIST | 1 | **0.73 ± 0.03** | 0.72 ± 0.02 | **0.70 ± 0.03** | 0.69 ± 0.02 | 0.72 ± 0.04 | **0.73 ± 0.02** | 0.70 ± 0.04 | **0.72 ± 0.04** |
| | 5 | **0.70 ± 0.03** | **0.70 ± 0.03** | **0.70 ± 0.02** | 0.70 ± 0.03 | **0.73 ± 0.02** | **0.73 ± 0.02** | **0.72 ± 0.02** | 0.69 ± 0.05 |
| | 10 | **0.70 ± 0.03** | **0.70 ± 0.03** | 0.70 ± 0.02 | **0.71 ± 0.03** | 0.72 ± 0.02 | **0.73 ± 0.02** | **0.72 ± 0.04** | 0.69 ± 0.05 |
| CIFAR-10 | 1 | **0.24 ± 0.02** | 0.23 ± 0.02 | **0.25 ± 0.03** | 0.24 ± 0.02 | **0.24 ± 0.04** | 0.22 ± 0.02 | **0.23 ± 0.03** | 0.21 ± 0.03 |
| | 5 | **0.23 ± 0.04** | 0.19 ± 0.03 | **0.21 ± 0.04** | 0.19 ± 0.02 | **0.21 ± 0.02** | 0.18 ± 0.02 | **0.22 ± 0.02** | 0.19 ± 0.04 |
| | 10 | **0.20 ± 0.02** | 0.18 ± 0.02 | **0.19 ± 0.03** | 0.19 ± 0.04 | **0.20 ± 0.03** | 0.16 ± 0.03 | **0.20 ± 0.03** | 0.17 ± 0.02 |
| IRIS | 1 | **0.95 ± 0.02** | **0.95 ± 0.02** | 0.95 ± 0.02 | **0.96 ± 0.02** | **0.95 ± 0.02** | **0.95 ± 0.02** | **0.97 ± 0.01** | 0.96 ± 0.01 |
| | 5 | **0.94 ± 0.02** | 0.86 ± 0.03 | **0.90 ± 0.02** | 0.83 ± 0.05 | **0.94 ± 0.02** | 0.86 ± 0.02 | **0.94 ± 0.03** | 0.86 ± 0.06 |
| | 10 | **0.85 ± 0.03** | 0.80 ± 0.03 | **0.80 ± 0.03** | 0.78 ± 0.02 | **0.84 ± 0.01** | 0.78 ± 0.03 | **0.80 ± 0.04** | 0.78 ± 0.03 |

## H  Dataset/Model Licenses and Sources

All datasets we use are publicly available. Our experiments include Mnist and SVHN datasets both distributed under the GNU General Public License. Reuters-21578 collection resides with Reuters Ltd. where Reuters Ltd. and Carnegie Group, Inc. allows free distribution of the dataset for research purposes only. Additionally, we utilize Emnist, Fashion-Mnist, Cifar-10, and Iris all with MIT License. We download Mnist, Reuters-21578, Fashion-Mnist, Cifar-10, and Iris from Keras datasets repository; Emnist from Tensorflow datasets repository, and SVHN from http://ufldl.stanford.edu/housenumbers/. LeNet, one of the models/neural networks we use, is distributed under MIT License whereas ResNet-50 is under Apache License 2.0.

## I  Ablation Study for Dependency Approximation under Top-$k$ Sampling

The primary motivation for choosing the Stochastic Softmax approximation is its computational simplicity. As shown in equation 14, this approximation enables the reuse of already computed acquisition scores without requiring additional operations. In contrast, the Stochastic Power method (equation 16) involves computing the logarithm of the scores, while the Stochastic Soft-rank method (equation 18) requires both sorting and logarithmic computation. These additional steps introduce extra computational overhead during query selection.

To assess the practical impact of this choice, we conducted empirical comparisons using these distributions to model the dependency under Top-$k$ selection. In Figure 10, we present results on the MNIST, Iris, Reuters, and Tiny-ImageNet datasets, which were selected to reflect varying data modalities, complexities,

Table 5: Comparison of mean classification test accuracies ±1 standard deviation for different sample selection sizes ($k$) during cycles and different sample selection strategies where number of samples in the labeled set $N_L = 100$. We use Least Confidence for the acquisition function and SSMS (Stochastic Softmax Sampling), SPS (Stochastic Power Sampling), SSRS (Stochastic Soft-rank Sampling), and Top-$k$ sampling for the sample selection strategy. The bold text highlights the higher mean accuracy and lower standard deviation when comparing DMLE and IMLE under various sampling schemes.

| Dataset | $k$ | SSMS | | SPS | | SSRS | | Top-$k$ | |
|---|---|---|---|---|---|---|---|---|---|
| | | DMLE | IMLE | DMLE | IMLE | DMLE | IMLE | DMLE | IMLE |
| MNIST | 1 | **0.80 ± 0.05** | 0.77 ± 0.03 | **0.78 ± 0.03** | **0.78 ± 0.03** | 0.78 ± 0.03 | **0.79 ± 0.03** | **0.78 ± 0.04** | 0.72 ± 0.04 |
| | 5 | **0.79 ± 0.03** | 0.77 ± 0.02 | **0.77 ± 0.04** | 0.76 ± 0.03 | **0.77 ± 0.02** | 0.74 ± 0.04 | **0.68 ± 0.06** | 0.66 ± 0.05 |
| | 10 | **0.77 ± 0.03** | **0.76 ± 0.03** | 0.76 ± 0.02 | **0.77 ± 0.03** | **0.75 ± 0.04** | 0.73 ± 0.05 | **0.62 ± 0.08** | 0.54 ± 0.07 |
| EMNIST | 1 | **0.35 ± 0.02** | 0.35 ± 0.03 | 0.37 ± 0.03 | **0.37 ± 0.01** | **0.37 ± 0.02** | 0.35 ± 0.04 | 0.31 ± 0.02 | **0.33 ± 0.02** |
| | 5 | **0.37 ± 0.04** | 0.36 ± 0.03 | **0.35 ± 0.03** | 0.34 ± 0.04 | **0.36 ± 0.04** | 0.35 ± 0.02 | **0.30 ± 0.03** | **0.30 ± 0.03** |
| | 10 | 0.36 ± 0.02 | **0.37 ± 0.03** | 0.36 ± 0.02 | **0.37 ± 0.03** | **0.35 ± 0.03** | 0.34 ± 0.04 | **0.27 ± 0.02** | **0.27 ± 0.02** |
| REUTERS | 1 | **0.50 ± 0.03** | 0.49 ± 0.02 | **0.49 ± 0.02** | 0.48 ± 0.03 | 0.48 ± 0.03 | **0.49 ± 0.02** | 0.40 ± 0.05 | **0.40 ± 0.02** |
| | 5 | **0.53 ± 0.02** | 0.51 ± 0.03 | 0.52 ± 0.03 | **0.52 ± 0.01** | **0.48 ± 0.03** | 0.46 ± 0.04 | **0.41 ± 0.07** | 0.39 ± 0.08 |
| | 10 | **0.51 ± 0.03** | 0.47 ± 0.04 | **0.50 ± 0.02** | 0.48 ± 0.02 | 0.44 ± 0.07 | **0.46 ± 0.04** | **0.33 ± 0.10** | 0.30 ± 0.09 |
| SVHN | 1 | **0.30 ± 0.04** | 0.28 ± 0.03 | **0.30 ± 0.04** | 0.29 ± 0.02 | **0.28 ± 0.03** | 0.23 ± 0.04 | 0.21 ± 0.03 | **0.23 ± 0.03** |
| | 5 | **0.25 ± 0.02** | 0.23 ± 0.04 | **0.24 ± 0.03** | 0.23 ± 0.03 | **0.22 ± 0.04** | 0.22 ± 0.04 | **0.20 ± 0.02** | 0.19 ± 0.04 |
| | 10 | 0.23 ± 0.03 | **0.23 ± 0.02** | **0.23 ± 0.03** | **0.23 ± 0.03** | **0.21 ± 0.04** | 0.20 ± 0.02 | **0.20 ± 0.02** | **0.20 ± 0.02** |
| FMNIST | 1 | **0.72 ± 0.01** | 0.72 ± 0.03 | 0.71 ± 0.02 | **0.73 ± 0.01** | 0.70 ± 0.04 | **0.71 ± 0.03** | **0.63 ± 0.04** | 0.62 ± 0.06 |
| | 5 | 0.71 ± 0.04 | **0.73 ± 0.01** | **0.72 ± 0.02** | 0.72 ± 0.04 | 0.71 ± 0.03 | **0.73 ± 0.04** | **0.56 ± 0.11** | 0.53 ± 0.06 |
| | 10 | **0.71 ± 0.04** | 0.70 ± 0.02 | **0.72 ± 0.03** | 0.71 ± 0.03 | 0.69 ± 0.04 | **0.70 ± 0.03** | **0.52 ± 0.06** | 0.51 ± 0.08 |
| CIFAR-10 | 1 | **0.23 ± 0.02** | 0.23 ± 0.03 | **0.24 ± 0.03** | 0.23 ± 0.02 | **0.22 ± 0.02** | 0.21 ± 0.02 | **0.20 ± 0.05** | 0.20 ± 0.03 |
| | 5 | **0.22 ± 0.04** | 0.18 ± 0.02 | **0.23 ± 0.04** | 0.20 ± 0.03 | **0.20 ± 0.02** | 0.19 ± 0.03 | **0.19 ± 0.02** | 0.18 ± 0.02 |
| | 10 | **0.21 ± 0.02** | 0.18 ± 0.03 | **0.21 ± 0.02** | 0.18 ± 0.02 | **0.17 ± 0.01** | 0.17 ± 0.03 | **0.18 ± 0.03** | 0.17 ± 0.03 |
| IRIS | 1 | 0.95 ± 0.02 | **0.96 ± 0.01** | **0.96 ± 0.02** | 0.95 ± 0.02 | **0.97 ± 0.01** | **0.97 ± 0.01** | **0.96 ± 0.01** | 0.95 ± 0.02 |
| | 5 | **0.94 ± 0.01** | 0.87 ± 0.02 | **0.94 ± 0.01** | 0.86 ± 0.03 | **0.94 ± 0.01** | 0.88 ± 0.04 | **0.91 ± 0.07** | 0.81 ± 0.06 |
| | 10 | **0.91 ± 0.03** | 0.80 ± 0.03 | **0.95 ± 0.02** | 0.80 ± 0.03 | **0.85 ± 0.03** | 0.79 ± 0.02 | **0.87 ± 0.06** | 0.76 ± 0.06 |

and sample selection sizes. The results show that the Stochastic Softmax consistently outperforms the other approximations in terms of average test accuracy under Top-$k$ sampling.

This empirical evidence supports the choice of a softmax-based approximation, highlighting its efficiency and effectiveness.

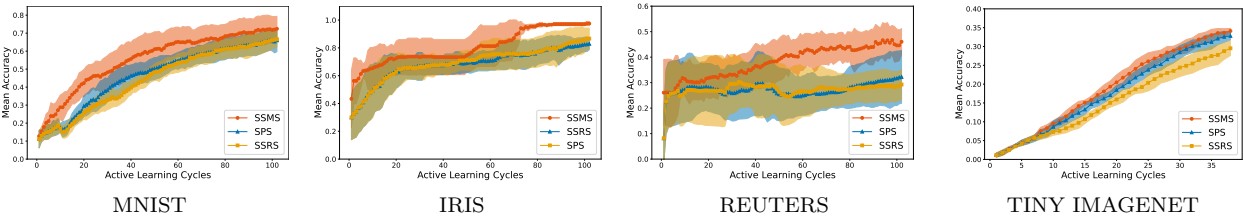

Figure 10: Comparison of average test accuracy (±1 standard deviation) using different distribution approximations for the dependency term with Top-$k$ selection: Stochastic Softmax (SSMS), Stochastic Power (SPS), and Stochastic Soft-rank (SSRS). These methods were evaluated during model parameter estimation on MNIST, Iris, Reuters, and Tiny ImageNet datasets. The results demonstrate the effect of each approximation across diverse data modalities and complexities, with sample selection sizes of $k = 1$ for the first three datasets and $k = 5$ for Tiny ImageNet.

## J   Glossary

This section presents a glossary of variables and notations used throughout the paper, provided in Table 6.

| Term / Symbol | Definition |
|---|---|
| $\mathcal{X} \subset \mathbb{R}^{d_x}$ | Input feature space of dimension $d_x$. |
| $\mathcal{Y} = \{1, \ldots, K\}$ | Label space with $K$ possible classes. |
| $x$ | Input sample from the feature space $\mathcal{X}$. |
| $y$ | Corresponding label of sample $x$, belonging to $\mathcal{Y}$. |
| $U_t = \{x_i\}_{i=1}^{N_U^t}$ | Unlabeled dataset at cycle $t$, containing $N_U^t$ samples. |
| $D_t = \{(x_i, y_i)\}_{i=1}^{N_L^t}$ | Labeled dataset at cycle $t$, containing $N_L^t$ labeled samples. |
| $N_U^t$ | Number of unlabeled samples at cycle $t$. |
| $N_L^t$ | Number of labeled samples at cycle $t$. |
| $S_{t+1}$ | Batch of samples selected for labeling at cycle $t$. |
| $\theta \in \mathbb{R}^{d_\theta}$ | Model parameters of dimension $d_\theta$. |
| $a(x, D, \theta)$ | Acquisition function assigning a score to an unlabeled sample $x$ based on labeled data $D$, model parameters $\theta$. |
| $f_\theta : \mathcal{X} \to \{0,1\}^K$ | Prediction function (e.g., deep neural network) parameterized by $\theta$, outputs scores for $K$ classes. |
| $t$ | Current cycle. |

Table 6: Glossary of terms and symbols used in the active learning framework.

