# OpenReview forum: "Dependency-aware Maximum Likelihood Estimation for Active Learning"
_TMLR — Accepted by TMLR_

### Review · Reviewer_Vmqk · 2025-04-12

**Summary Of Contributions:**

This paper studies an important issue in active learning, i.e., how to handle the example dependencies in the iteration process. Specifically, each sample acquisition influences subsequent selections, causing dependencies among samples in the labeled set. Existing methods ignore this issue. The author proposes DMLE to address sample dependencies and ensure consistency with active learning principles in the model parameter estimation process. The experimental results validate the effectiveness of the proposed approach.

Overall, I think this is a good article and tends to be accepted.

**Audience:**

Yes

**Claims And Evidence:**

Yes

**Requested Changes:**

N/A

**Strengths And Weaknesses:**

# Strengths
1. The problem studied in this paper is important and interesting in the active learning field.
2. The proposed approach is effective.
3. The toy experiment in Figure 1 is convincing.

# Weaknesses
1. In addition to the example dependency mentioned in this article, there is also a deep dependency between the selected example and the learned model. The selected samples are determined by the uncertainty of the model, and these samples further affect the updating of the model. Can you discuss these more?

---

> ### Author Response · Authors · 2025-05-21
>
> Thank you for the thoughtful and positive feedback, and we are glad that the contributions and motivation of our work were appreciated.
>
> We fully agree that the dependency between selected examples and the learned model, through uncertainty-based selection and its influence on model updates, is a key aspect of active learning. This is precisely the phenomenon that motivates our work. The DMLE approach is specifically designed to address the dependencies that arise from this iterative process and to ensure consistency in model parameter estimation within this feedback loop.

---

### Review · Reviewer_hMbu · 2025-05-08

**Summary Of Contributions:**

This submission introduces a novel approach, Dependency-aware Maximum Likelihood Estimation (DMLE), to address a fundamental mismatch between active learning and conventional model training techniques. Standard Maximum Likelihood Estimation (MLE) operates under the assumption that training samples are independently and identically distributed (i.i.d.), an assumption that fails in active learning where sample acquisition is sequential and each selection affects subsequent ones.

The key contributions are:

1. Identifying the MLE-i.i.d. mismatch in Active Learning
   The paper shows that conventional Independent MLE (IMLE) fails to account for dependencies introduced by sequential querying in active learning. This mismatch leads to inaccurate model updates and suboptimal performance in future acquisition cycles.

2. Proposing Dependency-aware MLE (DMLE)
   The authors propose DMLE, which corrects the training objective of MLE by explicitly incorporating the dependency structure of labeled data points across active learning cycles. The improved objective includes a correction term:

   $$
   \hat{\theta}_{\text{DMLE}} = \arg\max_\theta \sum_{(x,y) \in D_{t+1}} \log P(y \mid x; \theta) + \sum_{\tau=1}^{t+1} \sum_{x \in S_\tau} \log P(x \mid D_{\tau-1}; \theta)
   $$

   This formulation captures both label likelihood and feature-level sample selection probability, correcting the i.i.d. bias in IMLE.

3. Theoretical Analysis and Fisher Information Gain
   The authors show that DMLE leads to lower variance estimators than IMLE by proving:

   $$
   \mathcal{I}_{\text{DMLE}}(\theta) \succeq \mathcal{I}_{\text{IMLE}}(\theta) \Rightarrow \text{Cov}(\hat{\theta}_{\text{DMLE}}) \preceq \text{Cov}(\hat{\theta}_{\text{IMLE}})
   $$

   via the Cramér-Rao lower bound. This formalizes that DMLE yields more reliable parameter estimates.

4. General Applicability Across Acquisition Functions
   DMLE is shown to be agnostic to acquisition functions and sampling strategies, and can be plugged into any active learning pipeline as long as acquisition scores or ranks can be computed. The paper discusses entropy, BALD, Core-set, and least confidence as examples.

5. Empirical Validation Across Diverse Benchmarks
   Extensive experiments on 8 datasets (MNIST, FMNIST, SVHN, CIFAR-10, Reuters, Tiny ImageNet, Emnist, Iris) show that DMLE consistently outperforms IMLE across multiple acquisition functions (Entropy, BALD, Core-set) and sampling strategies (Top-k, Stochastic Softmax, Power, Soft-Rank Sampling). The gains are especially notable in early cycles, for example, average accuracy improvements of 6%, 8.6%, and 10.5% for batch sizes \( k = 1, 5, 10 \) respectively, after 100 acquired samples.

6. Statistical Significance and Robustness
   Using Wilcoxon signed-rank tests over multiple random seeds, the authors show that DMLE outperforms IMLE in 92% of the experiments, confirming the robustness and significance of the improvement.

**Audience:**

Yes

**Claims And Evidence:**

Yes

**Requested Changes:**

### Requested Changes

#### Critical

1. **Clarify the impact of the softmax approximation for Top-k sampling**
   The dependency term \( \log P(S_\tau \mid D_{\tau-1}; \theta) \) is approximated using a softmax-based distribution. However, the paper lacks analysis of how this approximation may affect theoretical guarantees or empirical results.
   **Action**: Include either a justification or empirical ablation (e.g., sensitivity to the approximation) to demonstrate the robustness of this assumption.

2. **Expand discussion on computational overhead**
   While DMLE introduces a theoretically grounded correction, it also incurs additional computational cost, especially in large-scale scenarios. The current discussion does not quantify the trade-off clearly.
   **Action**: Provide additional experimental details or complexity scaling plots that show the runtime/memory cost of DMLE versus IMLE under different settings.

3. **Improve notation clarity in Section 4**
   The presentation of the DMLE objective (particularly equations 9–11) can be dense for readers unfamiliar with active learning internals.
   **Action**: Define all variables precisely (e.g., clarify what each subscripted term like \( S_\tau \) and \( D_{\tau-1} \) refers to), and consider adding a small example or schematic.

#### Optional (but recommended to strengthen the paper)

4. **Include an ablation on acquisition functions and batch sizes**
   The paper reports performance across several acquisition functions and \( k \)-values, but could benefit from a focused comparison showing how DMLE performs relative to IMLE in particularly diverse or uncertain data regimes.
   **Suggestion**: Add a supplementary ablation plot or table showing accuracy improvements stratified by acquisition function difficulty.

5. **Discuss limitations and future directions more explicitly**
   The conclusion hints at future work, but a more structured discussion of DMLE's limitations (e.g., approximation assumptions, scalability, or diminishing returns in late cycles) would improve transparency.
   **Suggestion**: Add a short paragraph on when DMLE may not provide benefits, and how future work could address that.

6. **Real-world case study**
   The paper is well-validated on benchmarks, but its practical relevance could be enhanced with an example application in a high-stakes domain (e.g., medical, annotation-efficient language modeling).
   **Suggestion**: If space permits, add a discussion or a short real-world example, even if preliminary.

**Strengths And Weaknesses:**

### Strengths

1. **Novel Problem Formulation**
   The paper addresses a fundamental oversight in existing active learning literature: the incompatibility between the i.i.d. assumption of conventional Maximum Likelihood Estimation (MLE) and the sequential, dependent nature of data acquisition in active learning. This is a well-motivated and original contribution that fills a clear gap in methodology.

2. **Principled Theoretical Framework**
   The proposed Dependency-aware Maximum Likelihood Estimation (DMLE) is theoretically grounded. The authors derive the dependency term
   $$
   \sum_{\tau=1}^{t+1} \sum_{x \in S_\tau} \log P(x | D_{\tau-1}; \theta)
   $$
   and prove that incorporating it results in a Fisher Information gain, leading to lower-variance estimators according to the Cramér-Rao bound:
   $$
   \text{Cov}(\hat{\theta}_{\text{DMLE}}) \preceq \text{Cov}(\hat{\theta}_{\text{IMLE}})
   $$

3. **Generality and Practicality**
   DMLE can be applied with a variety of acquisition functions (entropy, BALD, Core-set) and selection strategies (Top-k, stochastic softmax, soft-rank, power sampling), making it widely usable in real-world active learning setups.

4. **Strong Empirical Results**
   The method demonstrates consistent and significant improvements across 8 diverse benchmark datasets, with test accuracy gains of up to 10.5% compared to IMLE. The experiments are thorough, covering multiple acquisition functions, sampling strategies, and batch sizes.

5. **Statistical Rigor**
   Results are backed by Wilcoxon signed-rank tests over multiple seeds, showing that DMLE significantly outperforms IMLE in 92% of settings, with p-values under 0.05.

6. **Reproducible and Clear Experimental Protocol**
   The authors detail their implementation setup, dataset splits, model choices, and training hyperparameters clearly, supporting reproducibility and transparency.

### Weaknesses

1. **Computational Overhead**
   Although DMLE adds meaningful improvements, the additional computation required to model the dependency term can increase training time. The paper could benefit from deeper discussion or mitigation strategies (e.g., approximations) for large-scale settings.

2. **Approximation of the Dependency Term**
   For Top-k sampling, the dependency term
   $$
   \log P(S_\tau | D_{\tau-1}; \theta)
   $$
   is approximated using a softmax distribution. While this is practical, the paper does not deeply analyze how this approximation affects theoretical guarantees or empirical results. A sensitivity analysis or ablation would strengthen the claims.

3. **Limited Discussion on Limitations**
   The paper could include a more critical reflection on scenarios where DMLE may not perform well—e.g., when the sample selection strategy already leads to highly diverse data, or when the model is overparameterized and less sensitive to data redundancy.

4. **No Real-world Application Case Study**
   While the synthetic and benchmark datasets demonstrate DMLE's benefits, applying it to a real-world active learning task (e.g., medical annotation or industrial settings) could enhance the practical relevance and highlight potential deployment considerations.

5. **Notation and Readability**
   Some formulas (particularly in Section 4) and variable definitions are dense and could benefit from clearer explanations or visual aids (e.g., a schematic of dependency modeling).

---

> ### Author Response · Authors · 2025-05-21
>
> Thank you for taking the time to review our work and provide detailed feedback! We have uploaded a revised version of the paper incorporating your requested and suggested changes, with all modifications highlighted in blue for clarity. Please refer to the latest revision to view the updates.
>
> **Critical**
>
> ***1. Clarify the impact of the softmax approximation for Top-k sampling (Added as Appendix I)***
>
> We appreciate the insightful comment regarding the use of the softmax-based approximation for the dependency term $\log P(S_\tau \mid D_{\tau-1}; \theta)$. This is indeed an important aspect, and we agree that its implications deserve further clarification.
>
> Our primary motivation for choosing the Stochastic Softmax approximation is its computational simplicity. As shown in Eq. 14, this approximation enables the reuse of already computed acquisition scores without requiring additional operations. In contrast, the Stochastic Power method (Eq. 16) involves computing the logarithm of the scores, while the Stochastic Soft-rank method (Eq. 18) requires both sorting and logarithmic computation. These additional steps introduce extra computational overhead during query selection.
>
> To assess the practical impact of this choice, we conducted empirical comparisons using these distributions to model the dependency under Top-$k$ selection. We have added Section I in the Appendix with Figure 10, presenting results on the MNIST, Iris, Reuters, and Tiny-ImageNet datasets, which were selected to reflect varying data modalities, complexities, and sample selection sizes. The results show that the Stochastic Softmax consistently outperforms the other approximations in terms of average test accuracy under Top-$k$ sampling.
>
> This empirical evidence supports our choice, demonstrating both the efficiency and effectiveness of the softmax-based approximation.
>
> ***2. Expand discussion on computational overhead (Added in Appendix F)***
>
> In Section 5.1 (Time Complexity), we present a theoretical comparison of the time complexities of DMLE and IMLE. To support this analysis empirically, Appendix F includes Table 5 comparing the mean elapsed times for both methods on the MNIST dataset with an initial pool of 30,000 unlabeled samples. The results, averaged over 8 runs per configuration with $k=1, 5, 10$, show that DMLE adds an average of only 1.88 minutes of runtime compared to IMLE when used with Stochastic Softmax Sampling (SSMS), Stochastic Power Sampling (SPS), and Stochastic Soft-rank Sampling (SSRS).
>
> As the difference between DMLE and IMLE lies in the objective function, we also plot the loss processing times per cycle to better observe the computational expense introduced by the dependency term in DMLE, which is added to Appendix F with Figure 9. This experiment, conducted on the MNIST dataset collecting 500 samples with sample selection size $k=1$, shows that the loss processing time increases for both methods across cycles as more samples are labeled. Although the difference between DMLE and IMLE grows over time, it remains relatively small in practice.
>
> We acknowledge, however, that as the labeled pool grows significantly, the computational burden of DMLE can increase due to its dependency-aware estimation. Nevertheless, within the context of active learning, where the goal is to achieve high performance with a limited number of labeled samples, we design our experiments to reflect this setting. Our results demonstrate that DMLE achieves competitive or superior performance using fewer labeled examples, thus mitigating the need for larger labeled sets and helping to keep the overall runtime within a practical and manageable range.
>
> ***3. Improve notation clarity in Section 4 (Added as Appendix J)***
>
> To improve clarity for readers who may be less familiar with the active learning process, we have added Figure 2, which illustrates the overall workflow of active learning and highlights the key terms used in our formulation, such as ( $S_\tau$ ) and ( $D_{\tau-1}$ ). Additionally, in response to your suggestion, we have included a glossary table (Table 6) and Appendix J, which define all relevant variables and notations. We hope these additions improve the readability and clarity of the paper.

---

> > ### Author Response · Authors · 2025-05-21
> >
> > **Optional**
> >
> > ***4. Include an ablation on acquisition functions and batch sizes***
> >
> > Although this type of analysis is not included in the current version, we see it as a promising direction for future work to better understand when and why DMLE outperforms IMLE, particularly in diverse or high-uncertainty data regimes. In particular, we plan to explore how DMLE interacts with model uncertainty and diversity-based selection strategies, which could lead to hybrid estimators that adaptively adjust dependency strength based on the phase of learning.
> >
> > ***5. Discuss limitations and future directions more explicitly (Updated Conclusion)***
> >
> > A promising direction for future work is the theoretical characterization of the dependency term's distribution under Top-$k$ selection strategies, beyond the current empirical analysis we provide. This would lead to deeper insights into the approximation behavior of DMLE and help identify scenarios where its benefits may diminish. Additionally, examining DMLE’s performance across varying uncertainty and diversity conditions represents a valuable line of research, with potential to inform improvements in the acquisition process. In response to this suggestion, we have updated the conclusion section to provide a more detailed discussion of potential future work and the exploration of DMLE’s limitations.
> >
> > ***6. Real-world case study***
> >
> > We agree that applying DMLE in a high-stakes domain such as medical decision-making would strengthen its practical relevance. While this paper focuses on demonstrating the effectiveness of DMLE across datasets with varying data and model complexities, exploring its impact in real-world medical settings is an exciting direction we are planning to pursue in future work. Due to the scope of this initial work, we have reserved the exploration of domain-specific applications for future studies.

---

### Review · Reviewer_b9XG · 2025-05-22

**Summary Of Contributions:**

This paper introduces a thoughtful approach to a subtle yet important aspect of the active learning process. Active learning aims to build an efficient labeled training set by strategically selecting samples for annotation. This paper focuses on the model parameter estimation stage within this iterative process.

The primary technical contribution of this work is the introduction of \emph{Dependency-aware Maximum Likelihood Estimation (DMLE)}, a novel method for model parameter estimation within the active learning framework.The authors identify a fundamental incompatibility: Active learning is an iterative process where each queried sample influences subsequent selections, thereby creating \emph{dependencies} among the samples in the labeled set. However, conventional MLE estimator, a standard for model updates, assumes that data samples are \textbf{independent and identically distributed (i.i.d.)}. This assumption is violated in the active learning context.

DMLE aims to rectify this by explicitly accounting for these dependencies. The standard MLE objective (which the paper refers to as Independent MLE or IMLE when the i.i.d. assumption is applied) is typically:
$\hat{\theta}_{t+1}^{\mathrm{IMLE}} = \argmax_{\theta} \sum_{(x,y) \in D_{t+1}} \log P(y|x;\theta) $


This formulation arises from a more complete likelihood expression that includes a term for the probability of selecting the samples themselves:
$ \hat{\theta}_{t+1}^{\mathrm{MLE}} = \argmax_{\theta} \left( \sum_{(x,y) \in D_{t+1}} \ln P(y|x;\theta) + \sum_{\tau=1}^{t+1} \ln P(S_{\tau}|D_{\tau-1};\theta) \right) $

The term $P(S_{\tau}|D_{\tau-1};\theta)$ represents the probability of selecting the batch of samples $S_{\tau}$ given the previously labeled data $D_{\tau-1}$ and model parameters $\theta$. Under the i.i.d. assumption, this term is considered constant and thus dropped.

The technical advancement of DMLE lies in retaining and modeling this dependency term. The resulting DMLE objective function is:
$$ \hat{\theta}_{t+1}^{\mathrm{DMLE}} = \argmax{\theta} \left( \sum_{(x,y) \in D_{t+1}} \ln P(y|x;\theta) + \sum_{\tau=1}^{t+1} \sum_{x \in S_{\tau}} \ln P(x|D_{\tau-1};\theta) \right) $$

A key challenge is that the distribution $P(S_{\tau}|D_{\tau-1};\theta)$ can be difficult to express in a closed form. The authors cleverly address this by leveraging stochastic batch sampling strategies (like Softmax, Power, and Soft-Rank acquisitions as proposed by Kirsch et al., 2021), which have defined probability distributions for sample selection. For the common Top-k selection strategy, DMLE approximates this probability using the Softmax distribution.

The authors provide two theorems as theoretical support for DMLE. The first one posits that DMLE achieves a Fisher Information matrix $I^{\text{DMLE}}(\theta)$ that is greater than or equal to that of IMLE ($I^{\text{IMLE}}(\theta)$). According to the Cramér-Rao bound, this implies that DMLE can provide parameter estimates with lower variance ($\text{Cov}(\hat{\theta}^{\text{DMLE}}) \le \text{Cov}(\hat{\theta}^{\text{IMLE}})$), leading to enhanced precision. The second one demonstrates that maximizing the expected log-likelihood with DMLE is equivalent to minimizing the sum of two Kullback-Leibler (KL) divergences. One KL divergence measures the difference between the true conditional label distribution and the model's estimate, while the second (the dependency term often ignored by IMLE) measures the discrepancy between the true data distribution $P(x')$ and the model's estimate of it based on actively selected data $P(x'|D_t; \hat{\theta})$. This ensures the model aligns better with both the predictive and the underlying data distributions.

The paper empirically evaluates DMLE against IMLE across several benchmark datasets, including image (MNIST, Fashion-MNIST, SVHN, EMNIST, Cifar-10, Tiny ImageNet), text (Reuters), and feature-based (Iris) datasets. Various acquisition functions (entropy, Core-set, BALD, least confidence) and sample selection strategies (Stochastic Softmax, Stochastic Power, Stochastic Soft-rank, Top-k) are employed.

**Audience:**

Yes

**Claims And Evidence:**

Yes

**Requested Changes:**

Please address the weakness I mentioned above.

**Strengths And Weaknesses:**

S1. Well-formulated problem from a specific MLE-aware perspective of active learning
S2. Well-established theorems to provide deep insight on the problem and the author's solution (pls refer to the summary of contributions)
S3. Extensive evaluation to understand the effectiveness of proposed algorithms under various controllable simulation with real-world datasets.

My major concern is the relevance of this work to the lanscape of current machine learning research. The concern regarding the applicability of batch-based active learning with the reported experimental setup to the frontiers of the domain is quite pertinent.

W1. The experiments, while covering a variety of data types, primarily utilize datasets (e.g., MNIST, Iris, Cifar-10) and network architectures (e.g., LeNet, small MLPs) that, while standard for benchmarking certain algorithms, are modest in scale compared to the massive datasets and extremely large models that characterize much of cutting-edge AI research today. The ResNet-50 and ViT-B/16 are more contemporary, but the overall experimental scope might not fully capture the dynamics at the current AI frontier.

W2. The query batch sizes ($k=1, 5, 10$) are relatively small. In large-scale applications, the cost of retraining or fine-tuning enormous models might necessitate much larger batch queries to be practical, or different paradigms altogether.

W3. While the paper analyzes time complexity and reports manageable additional costs for DMLE on the tested scales, the computational implications of the dependency term, especially the calculation or approximation of $P(x|D_{\tau-1};\theta)$ and the additional summation in the loss, could become more pronounced with vastly larger datasets and more complex models. The authors do simplify the normalization constant $Z_{\tau,i}$ based on empirical observations on MNIST to manage computational expenses.

I am considering leveraging a pre-trained VLM combined with prompt engineering as a method to efficiently generate high-quality annotations for all unlabeled images. This approach could significantly streamline the annotation process while maintaining accuracy. Additionally, models like CLIP could be utilized to assign pseudo-labels to the unlabeled data. These pseudo-labels can serve as valuable inputs for supervised or semi-supervised machine learning tasks, potentially enhancing overall model performance and achieving better outcomes in downstream applications.

After all, the concept of active learning was originally founded on the premise that high-quality data annotation is both time-consuming and expensive, making it essential to carefully select an optimal subset of data for annotation. In the current machine learning landscape, however, this challenge has likely evolved to a more complex level. Relying on traditional batch-based sampling methods applied to relatively small and straightforward classification datasets may no longer be sufficient to address the demands of modern, large-scale, and diverse datasets. You should consider something that the pre-trained models (which are almost free) cannot do.

---

> ### Author Response · Authors · 2025-05-27
>
> Thank you for your thoughtful and detailed feedback! We are especially grateful for the recognition of our main contributions, including the identification of the incompatibility between the i.i.d. assumption and active learning sample dependencies, the introduction of DMLE to explicitly account for these dependencies, and the offering of theoretical guarantees alongside extensive empirical validation across diverse datasets and acquisition functions.
>
> We respectfully address the reviewer’s main concerns below.
>
> ***W1: The experiments primarily use small-scale datasets and modest architectures***
>
> Our choice of datasets and models was deliberate. Benchmarks such as MNIST, CIFAR-10, and Fashion-MNIST, while considered small-scale, remain widely used in active learning studies, including in recent top-tier papers, precisely because they enable interpretable and reproducible experimentation. Accordingly, we explore the impact of sample dependencies on model parameter estimation using widely adopted datasets to ensure both relevance and clarity.
>
> We also included more modern architectures, such as ResNet-50 and ViT-B/16, and larger datasets like Tiny ImageNet and Reuters to demonstrate that our findings extend beyond toy settings. Our contribution is methodological and theoretical in nature, focusing on the core mechanics of training under sample dependencies rather than proposing new architectures or achieving state-of-the-art performance. Ensuring compatibility between the estimator and the nature of the training data, particularly when the i.i.d. assumption is violated, is foundational and broadly applicable across different scales and architectures.
>
> ***W2: Small query batch sizes may not reflect the realities of large-scale applications***
>
> While we follow common active learning practices that assume high annotation costs and therefore use small batch sizes, our method is not restricted to such settings, as we show in our results. The dependency-aware correction introduced by DMLE is applicable regardless of the number of queried samples and scales algorithmically to larger batches. In fact, we observe that the importance of using DMLE increases with larger selection sizes, as shown in Tables 2, 3, and 4, since the accumulation of dependencies between samples becomes more pronounced.
>
> Our use of small batches is intended to make these effects more observable and measurable in a controlled, iterative fashion. However, our empirical findings and theoretical formulation both support the utility of DMLE in settings with larger batch selections.
>
> ***W3: Computational cost may be a concern at scale***
>
> We agree that scalability is important. As discussed in our response to Reviewer #hMbu and further detailed in Appendix F, we provide both theoretical and empirical analyses of the runtime overhead introduced by DMLE. Although DMLE adds a modest increase in training time due to its dependency-aware objective, this overhead remains manageable. Furthermore, the additional cost is offset by DMLE’s improved sample efficiency, which reduces the number of labeled examples required and helps keep the overall training time practical within the active learning framework.
>
> ***Response to concerns with relevance of active learning***
>
> We view DMLE as complementary to approaches that use pseudo-labeling or prompt-based vision-language models. While such methods are increasingly adopted, they still rely on model parameter estimation, and our work strengthens this core component. As data selection becomes more selective or biased, even in large-scale or hybrid pipelines, ensuring that model training remains consistent with how data is acquired becomes increasingly critical.
>
> *We appreciate the reviewer’s suggestion to consider pre-trained models and prompt-based methods. However, we believe these directions are orthogonal to our contribution. DMLE is not intended to replace zero-shot or few-shot learning, but rather to improve the statistical consistency of active learning itself. Our work addresses a subtle but foundational issue: the mismatch between data selection strategies and standard training assumptions. This remains relevant regardless of whether annotations come from human labelers, model predictions, or hybrid approaches*.

---

> ### Comment · Reviewer_b9XG · 2025-05-27
> **Good response**
>
> Upon initial review, your response is substantially correct. Your theoretical insights should offer a perspective that is independent of, and can enhance, current engineering-oriented practices.I really appreciate your efforts actually.
>
> For large batch selection in active learning, I suggest referring to Andreas Krause's earlier work ([1] and others). His pioneering research utilized submodularity to surrogate the utility (such as mutual information or uncertainty reduction based on GP) of subset selection from large datasets. You might consider discussing the potential of adapting your DMLE framework to submodular functions for active learning. In this way, you could figure out a promising direction for future research focused on selecting larger batches through combinatorial methods and your proposed objective. Furthermore, submodularity problem can be almost solved by greedy or its derivatives subject to varying optimization constraints. In this way, you could keep your focus on complexity trade-off.
>
> [1] Adaptive submodularity: Theory and applications in active learning and stochastic optimization D Golovin, A Krause - Journal of Artificial Intelligence Research, 2011

---

> > ### Author Response · Authors · 2025-05-27
> >
> > Thank you for your thoughtful follow-up and for highlighting the connection to submodular optimization. We appreciate the pointer to Krause’s work on adaptive submodularity, which indeed offers a principled foundation for large-batch selection in active learning. This perspective aligns well with our long-term goal of scaling DMLE beyond small batch settings.
> >
> > While our current focus is on correcting the mismatch between training objectives and selection-induced dependencies, we agree that combining our dependency-aware estimation with submodular selection strategies is a promising direction. Such integration could enhance both the theoretical foundation and scalability of active learning systems that employ DMLE for model parameter estimation.
> >
> > Please check the revised conclusion section, where we have included this possible future direction. We appreciate your suggestion and your encouraging remarks.

---

### Decision · Action_Editor_twt1 · 2025-08-21

**Recommendation:** Accept as is

**Audience:**

Yes

**Audience Explanation:**

This paper studies active learning (adaptively requesting labels to minimizing labeling effort), which should be of interest to the TMLR community.

**Claims And Evidence:**

Yes

**Claims Explanation:**

The paper formulates a new optimization objective for maximum likelihood-based active learning, and demonstrates through extensive experiments that optimizing the proposed objective (DMLE) trains better models than optimizing the conventional objective (IMLE). All reviewers rate this paper positively so I support their recommendations.

In communication with the authors, I see that
- The DMLE algorithm assumes that $S_{\tau+1} | D_\tau, U_\tau$ was drawn using the scoring of $\theta$.
- In the actual active learning process, $S_{\tau+1} | D_\tau, U_\tau$ was drawn using the scoring of $\hat{\theta}_\tau$.

So I actually think there is some model mismatch in DMLE's likelihood function definition. This is in line with e.g. Eq. (11) of [1] or [2, Chap 15] (and possibly every interactive learning lower bound paper where they explicitly characterize the sequential data generation process), where only the first part ($\prod_{t=1}^n P_{Y|X}^m (y_t | x_t)$) depends on the regression function ($f^m$). I encourage the authors to discuss this in the final version.

[1] Raginsky and Rakhlin, "Lower bounds for Passive and Active Learning", NeurIPS 2011.
[2] Lattimore and Szepesvari, "Bandit Algorithms", 2020.

---

> ### Author Response · Authors · 2025-09-19
> **Camera-ready submission**
>
> We appreciate the timely review process and decision. We have incorporated the suggested discussion and uploaded the camera-ready version along with the GitHub link to the code. We thank the reviewers and editors for their positive feedback and constructive suggestions, which helped improve the paper.